



# Level 2 processor and auxiliary data for ESA Version 8 final full mission analysis of MIPAS measurements on ENVISAT

Piera Raspollini[1], Enrico Arnone[2, 3], Flavio Barbara[1], Massimo Bianchini[4], Bruno Carli[1], Simone Ceccherini[1], Martyn P. Chipperfield[5,6], Angelika Dehn[7], Stefano Della Fera[1,8], Bianca Maria Dinelli[2], Anu Dudhia[9], Jean-Marie Flaud[10, 18], Marco Gai[1], Michael Kiefer[11], Manuel López-Puertas[12], David P. Moore[13, 14], Alessandro Piro[15], John J. Remedios[13, 14], Marco Ridolfi[16], Harjinder Sembhi[14], Luca Sgheri[17], and Nicola Zoppetti[1]

[1]Istituto di Fisica Applicata Nello Carrara, Consiglio Nazionale delle Ricerche (IFAC-CNR), I-50019 Sesto Fiorentino, Italy
[2]Istituto di Scienze dell'Atmosfera e del Clima, , Consiglio Nazionale delle Ricerche (ISAC-CNR), I-40129 Bologna, Italy
[3]Department of Physics, University of Turin, 10125 Torino, Italy
[4]Istituto dei Sistemi Complessi (ISC) del Consiglio Nazionale delle Ricerche (CNR), Section of Florence, Italy
[5]School of Earth and Environment, University of Leeds, Leeds, LS2 9JT, UK
[6]National Centre for Earth Observation (NCEO), University of Leeds, Leeds, LS2 9JT, UK
[7]ESA-ESRIN, Frascati, Italy
[8]Department of Physics and Astronomy, University of Bologna, Bologna, Italy
[9] Department of Physics, University of Oxford, UK
[10]Laboratoire Interuniversitaire des Systèmes Atmosphériques (LISA), UMR CNRS 7583, Université de Paris, Université Paris-Est Créteil
[11]Karlsruhe Institute of Technology, Institute of Meteorology and Climate Research, Karlsruhe, Germany
[12]Instituto de Astrofísica de Andalucía, CSIC, Granada, Spain
[13]National Centre for Earth Observation (NCEO), University of Leicester, Leicester, United Kingdom,
[14]School of Physics and Astronomy, University of Leicester, Leicester, United Kingdom
[15]SERCO SpA c/o European Space Agency ESA-ESRIN, Frascati, Italy
[16]Istituto Nazionale di Ottica, Consiglio Nazionale delle Ricerche (INO-CNR), I-50019 Sesto Fiorentino, Italy
[17]Istituto per le Applicazioni del Calcolo, Consiglio Nazionale delle Ricerche (IAC-CNR), Section of Florence, Italy
[18]Institut Pierre-Simon Laplace, 61 avenue du Général de Gaulle, 94010 Créteil Cedex, France

**Correspondence:** Piera Raspollini (p.raspollini@ifac.cnr.it), Marco Gai (m.gai@ifac.cnr.it)

**Abstract.** High quality long-term data sets of altitude-resolved measurements of the atmospheric composition are important because they can be used both to study the evolution of the atmosphere and as a benchmark for future missions. For the final ESA reprocessing of MIPAS (Michelson Interferometer for Passive Atmospheric Sounding) on ENVISAT (ENViromental SATellite) data, numerous improvements were implemented in the level 2 (L2) processor Optimised Retrieval Model (ORM)

version 8.22 (V8) and its auxiliary data. The implemented changes involve all aspects of the processing chain, from the modelling of the measurements with the handling of the horizontal inhomogeneities along the line of sight to the use of the Optimal Estimation for retrieving the minor species, from a more sensitive approach to detect the spectra affected by clouds to a refined method for identifying low quality products. Improvements in the modelling of the measurements were obtained also with an update of the used spectroscopic data and of the databases providing the a priori knowledge of the atmosphere. The

HITRAN_mipas_pf4.45 spectroscopic database was finalised with new spectroscopic data verified with MIPAS measurements themselves, while recently measured cross-sections were used for the heavy molecules. The so-called IG2 data set, containing





the climatology used by MIPAS L2 processor to generate the initial guess and interfering species profiles when the retrieved profiles from previous scans are not available, was improved taking into account the diurnal variation of the profiles defined using climatologies from both measurements and models. Horizontal gradients were generated using ECMWF ERA-Interim

data closest in time and space to the MIPAS data. Further improvements in the L2 V8 products derived from the use of the L1b V8 products, which were upgraded to reduce the instrumental temporal drift and to handle the abrupt changes in the calibration gain. The improvements introduced into the ORM V8 L2 processor and its upgraded auxiliary data, together with the use of the L1b V8 products, lead to the generation of the MIPAS L2 V8 products that are characterised by an increased accuracy, better temporal stability, and a greater number of retrieved species.

## 1 Introduction

Atmospheric composition is changing due to the anthropogenic emissions of greenhouse gases and pollutants, to the reduction of the ozone-depleting substances regulated by the Montreal Protocol and to natural variability, including volcanic eruptions and solar activity. These changes affect the whole atmosphere from the surface to the mesosphere and largely vary in time, altitude and latitude. High quality long-term data sets of global altitude-resolved atmosphere composition measurements are

essential to understand the interaction between the changes in atmospheric composition and circulation and their impact on climate, weather (Kidston et al., 2015) and air quality. Furthermore, they can be used as a benchmark for future missions.

MIPAS (Michelson Interferometer for Passive Atmospheric Sounding) is a Fourier transform spectrometer for the measurement of the atmospheric composition at altitudes from the upper troposphere to the thermosphere, with a special focus on the stratosphere. It is one of few instruments that allowed vertically resolved sounding of the atmospheric emission over a long

period. It operated on board the Envisat satellite in a Sun-synchronous polar orbit for 10 years, from July 2002 to April 2012. It observed the atmospheric emission at the limb in the middle infrared spectral region, allowing continuous measurements during both day and night. The sequence of limb observations at different tangent altitudes provides information about the concentration of the emitting constituents as a function of altitude at high vertical resolution. Middle infrared spectra contain features of numerous species: $CO_2$, used for temperature retrieval, water vapour, ozone and many other longer-lived green-house gases,

species of interest for ozone-chemistry, many nitrogen and sulphur compounds, gases produced by biomass burning and other pollution plumes, as well as some isotopologues.

The analysis of MIPAS measurements is performed in two steps: from the interferograms measured by the instrument, the Level 1b (L1b) analysis (Kleinert et al. (2007, 2018)) produces the geolocated and radiometrically calibrated spectra. These are then injected into the Level 2 (L2) processor, which, starting from these spectra, retrieves the concentration of the atmospheric

parameters of interest. The inversion procedure is based on the simulation (with a full-physics model) of the atmospheric emission spectra measured by the instrument (computed assuming known atmospheric parameters and information on the instrument response) and on the determination of the atmospheric parameters which minimise the differences between the modelled observations and the real observations. The quality of MIPAS L2 products depends upon the quality of the L1b products and on the accuracy of the L2 processor in modelling the observations, taking care that all systematic errors are



minimised. The activities related to the improvements of the L1b and L2 analyses advanced in parallel during the MIPAS mission, but with large cross-fertilization between them: detected anomalies in the L2 products motivated investigations and improvements on the L1b products, while changes in the L1b products were verified also looking at their impact on the L2 products. ESA (European Space Agency) L2 processor, based on the Optimised Retrieval Model (ORM) algorithm described in Ridolfi et al. (2000) and Raspollini et al. (2006), was originally designed to perform the Near-Real-Time (NRT) analysis

of the MIPAS measurements, with the requirement of a processing time of less than three hours to allow assimilation of the products by ECMWF (European Centre for Medium-Range Weather Forecasts) (Dragani, 2012; Thépaut et al., 2012).

The end of the mission did not stop the work for improving the data and two full mission reprocessings were performed, one with the MIPAS Level 2 Processing Prototype (ML2PP) code Version 6 (V6) (Raspollini et al. (2013)) using L1b V5 data, the second with ML2PP V7 (De Laurentis et al. (2016); Valeri et al. (2017)) with L1b V7 data. These ESA reprocessings were

performed with improvements of the original NRT algorithm, but other independent L2 reprocessing were also performed (Dinelli et al. (2010); Dudhia (2008); Kiefer et al. (2021)).

For the final reprocessing of the whole MIPAS mission a significant effort was made by the MIPAS Quality Working Group, supported by ESA, to further improve both L1b and L2 processors, as well as spectroscopy and a priori knowledge of the atmosphere. The objective was to obtain L2 products with increased accuracy, reduced instrumental temporal drift, reduced

discontinuities in the timeseries, and a greater number of retrieved species.

The improvements implemented in the L1b V8 processor are described in Kleinert et al. (2018), where the error estimate of the L1b products is also provided. Here we focus on the description of the main features and recent changes implemented in ORM version 8.22 (used for the final full mission reprocessing and referred in the rest of the paper as ORM V8 or ESA L2 processor V8), which also includes the use of L1b V8 products. Each implemented improvement is discussed by analysing its

impact on the quality of L2 products. Some of the improvements implemented in the L1b processor are also briefly recalled with the intent of highlighting their direct impact on the quality of the L2 products. The description of the implemented improvements is meant to better understand the products of this processor and to inspire future developers of retrieval codes.

The paper is organized as follows. In Sect. 2 the MIPAS on ENVISAT mission and the characteristics of the measurements are briefly recalled, while in Sect. 3 the main characteristics of the retrieval program implemented in L2 processor and its

auxiliary data are discussed. The subsequent three sections are dedicated to the description of the improvements in the L2 analysis, in particular Sect. 4 is dedicated to the changes in the modelling of the measurements, including the use of an updated spectroscopic database and of the state-of-the-art knowledge of the atmosphere for the definition of initial guess and a priori profiles. Sect. 5 deals with changes that make possible the retrieval of very minor species, Sect. 6 with the choices adopted to reduce the number of outliers in the products. Sect. 7 describes the impact of improvements implemented in L1b V8 data

on the reduction of both the instrument drift and the error in the radiometric calibration, Sect. 8 deals with the changes in the format of the output files. Finally, the conclusions are given in Sec. 9.



## 2 MIPAS on ENVISAT mission

MIPAS (Fischer et al., 2008) is a Fourier transform spectrometer that measured atmospheric limb emission spectra on board the ENVISAT satellite launched by ESA in 2002. Flying at about $800 \ \mathrm{km}$ with a near-circular polar sun-synchronous orbit
inclined of $98.55°$ with respect to the plane of the equator, it overpassed each region on Earth at the constant local time of 10 hour, with the dayside measurements being performed during the descending part of the orbit (from North to South) and the nightside measurements during the ascending one.

MIPAS was installed at the rear of Envisat, looking backwards with respect to the satellite's flight direction. Only around the poles, in order to get a better latitude coverage, the line of sight's azimuth was commanded to the poleward side of the flight
path. A very small fraction of the measurements were performed looking sideways on the side opposite to the sun to explore the possibility of aircraft emission measurements.

With an orbit period of 100.6 minutes, 14.3 orbits per day were performed. The quasi-polar orbit allowed a global coverage, with a revisit time of three days. The instrument has two input ports (one receives the radiation from the atmosphere and the other is designed to look at a cold target in order to minimize its contribution to the energy load on the detectors) and two output
ports, each of them equipped with four detectors (A1 to D1 and A2 to D2 for the two ports, respectively) centered at different wavenumbers that together cover the spectral range from 685 to $2410 \ \mathrm{cm}^{-1}$. The spectra from the eight detectors are combined in five spectral bands (denoted A ($685\text{-}980 \ \mathrm{cm}^{-1}$), AB ($1010\text{-}1180 \ \mathrm{cm}^{-1}$), B ($1205\text{-}1510 \ \mathrm{cm}^{-1}$), C ($1560\text{-}1760 \ \mathrm{cm}^{-1}$), and D ($1810\text{-}2410 \ \mathrm{cm}^{-1}$)) in the L1b products. The long-wavelength channels A1, A2, B1, and B2 use photoconductive mercury cadmium telluride (MCT) detectors, while photovoltaic MCT detectors are used in the short wavelength channels C1, C2, D1,
and D2.

In the first 2 years of operation (from July 2002 to March 2004), MIPAS acquired, nearly continuously, measurements at full spectral resolution (FR), with maximum path difference of $20 \ \mathrm{cm}$, corresponding to a spectral sampling of $0.025 \ \mathrm{cm}^{-1}$. A spectrum at the maximum spectral resolution is measured in $4.5 \ \mathrm{sec}$. On 26 March 2004, FR measurements were discontinued due to a mechanical problem in the interferometer drive unit. Atmospheric measurements were resumed in January 2005, with
a maximum path difference reduced to $8 \ \mathrm{cm}$, corresponding to a spectral sampling of $0.0625 \ \mathrm{cm}^{-1}$ and a measurement time of each spectrum of $1.8 \ \mathrm{sec}$. The reduced measurement time was exploited to acquire a larger number of spectra for each scan and hence a finer vertical sampling could be implemented. This turned out to be a better compromise between spectral resolution and vertical resolution. The advantage of high spectral resolution is a reduction of spectral interferences and a better characterisation of the spectra, while the advantage of a high vertical resolution is a better monitoring of the constituent
distributions. Having already obtained information on the spectral characterisation in the first part of the mission, for the long-term operations an improvement of both vertical and latitude resolution was considered to be a good compromise. For this, the measurements acquired from January 2005 onward are referred to as optimized resolution (OR) measurements. The nominal FR (OR) scan pattern consists of 17 (27) sweeps with tangent heights in the range from 6 to 68 (7–72) km with 3 (1.5) km steps in the upper troposphere-lower stratosphere (UTLS) region. Further than the nominal measurement modes described above,
which are used for most of MIPAS measurements in both phases of the mission, other measurement modes were acquired





and processed. Some are focused on UTLS region, some on the Middle Atmosphere, some on the Upper Atmosphere. Both horizontal and vertical sampling varies for the different measurement modes. The horizontal sampling varies from 550 km in the first phase of the mission to about 410 km in the second phase for the nominal mode, but there are measurements modes where it is even smaller. Further details of the characteristics of the MIPAS measurement modes are contained in (Dinelli et al., 2021) and reference therein. Almost all measurement modes are processed by the ESA processor.

## 3   Review of theoretical baseline of the L2 algorithm

The ORM Level 2 processor (Ridolfi et al., 2000; Raspollini et al., 2006, 2013) was specifically designed to operate in NRT and to use a minimum amount of a priori information that may introduce a bias in the profiles. To this end, the altitude grid of the retrieval coincides with the tangent points of the limb measurements (or a subset of them) where the sensitivity of the measurements peaks. The retrieval is performed using the global fit method (Carlotti, 1988), consisting in the simultaneous fit of the whole limb scanning sequence of the spectra acquired at different tangent altitudes. The non-linear least-square fit is used and the chi-square is minimized using the Gauss-Newton approach with an iterative procedure. The ill-conditioned problem of the measurements is handled with the regularizing Levenberg–Marquardt approach (Levenberg, 1944; Marquardt, 1963; Hanke, 1997; Doicu et al., 2010) during the iterations and with an a posteriori regularization with a self-adapting constraint dependent on the random error of each profile (Ceccherini, 2005; Ridolfi and Sgheri, 2011) applied after convergence. An accurate method, specifically designed for the regularizing Levenberg–Marquardt approach, is used for the computation of the diagnostic quantities (covariance matrix (CM) and averaging kernel matrix (AKM)) (Ceccherini and Ridolfi, 2010). The forward model internal to the retrieval code simulates the atmospheric radiance measured by the spectrometer as the result of the radiative transfer in a non-uniform medium. In the early versions the medium was non-uniform in the vertical direction, and, since V8 of the processor (see Sect. 4.1), is non-uniform in the horizontal direction as well. Instrument effects are also taken into account. The Instrument Line Shape (ILS) and the Instantaneous Field of View (IFOV) are modelled using an accurate instrument characterisation (Fischer et al., 2008). Scattering is not included in the radiative transfer integral, and the spectra affected by thick clouds, identified by a cloud filtering algorithm (Spang et al., 2002, 2004; Raspollini et al., 2006), are not considered in the analysis.

The atmosphere is assumed to be in local thermodynamic equilibrium (LTE) and in hydrostatic equilibrium. The impact of unaccounted atmospheric effects (non-LTE, interfering species, etc.) is minimized through the selection of spectral intervals (microwindows) containing relevant information on target parameters and minimizing the systematic errors ((Dudhia et al., 2002; Dudhia, 2021)). Furthermore, retrievals are performed only up to 78 km as a maximum. Other algorithms which take Non-LTE into account (see e.g. López-Puertas et al., 2018, and references therein) perform the analysis in the whole mesosphere.

Mutual interference of species that contribute to the same spectral region is handled with individual constituents retrieved according to an order of spectroscopic relevance. In this way the main interfering species are modelled with a concentration that has been previously derived from the same atmospheric sample. The only exception to the sequential retrieval is the case





of the pressure corresponding to the tangent altitudes and the related temperature values (pT retrieval). These two quantities

are retrieved simultaneously exploiting the external information provided by the hydrostatic equilibrium and the engineering knowledge of the limb scanning steps. Therefore, for each scan, the first operation is the simultaneous pT retrieval, followed by a sequential retrieval of trace gases volume mixing ratio (VMR) profiles (first $H_2O$, then $O_3$, and all the other species). The retrieval vector includes, in addition to the species (or pT) profile, microwindow-dependent continuum transmission profiles and microwindow-dependent but height-independent offset calibration values. The improvements implemented in the L2 processor

ORM V8 involve both the forward and inverse models, as well as the approach for filtering out spectra affected by clouds and for filtering out profiles with bad quality.

## 4 Improvements in the modelling of the measurements

As stated above, the retrieval of atmospheric parameters from MIPAS remote sensing measurements requires the fit of the simulated measurements to the observations through an iterative procedure. The fit is performed through the minimization of

the chi-square function, defined as the weighted $L^2$ norm of the residuals (given by the difference between the observations and the simulations), with the weight given by the inverse of the CM of the observations. The chi-square function is then normalized with the number of degrees of freedom of the fit (i.e. the number of observations minus the number of the retrieved parameters). If Optimal Estimation (see Sect. 5.1) is used, the function to be minimized takes into account also the constraints imposed to the retrieval, namely the square differences between the retrieved parameters and the a priori parameters, weighted

by the corresponding a priori error.

Any effort in improving the modelling of the measurements helps in reducing the systematic errors and this leads to a better accuracy of the products. These improvements focused on three main aspects:

– the handling of the horizontal inhomogeneities along the line of sight in the forward model,

– the use of more accurate spectroscopic parameters and cross-sections for heavy molecules,

– the use of the state-of-the-art representation of the atmosphere.

The overall impact of these modifications on the modelling of the observations can be evaluated in terms of the reduction of the chi-square with respect to the previous processing version. Figures 1 and 2 compare the chi-square histograms of some representative trace gas retrievals obtained from V7 and V8 L2 data sets, for MIPAS FR and OR nominal measurements respectively. Most of the histograms show a double peak that corresponds to the different impact of interference of non-target

gases in the various latitude bands explored by the measurements along the orbits. For most of the retrieved gases we observe a reduction in the V8 chi-square as compared to V7. The reduction is more significant in the OR phase of the mission, where the reduced spectral resolution makes more critical the interference of non target gases. The retrievals characterized by the largest reduction in the chi-square are $H_2O$ (-8%), $O_3$ (-20%), $HNO_3$ (-5%), $N_2O$ (-5%), $NO_2$ (-7%), $CFC-12$ (-6%), $N_2O_5$ (-9%), $ClONO_2$ (-14%). According to the results of dedicated sensitivity tests (not presented here), the obtained chi-square reductions





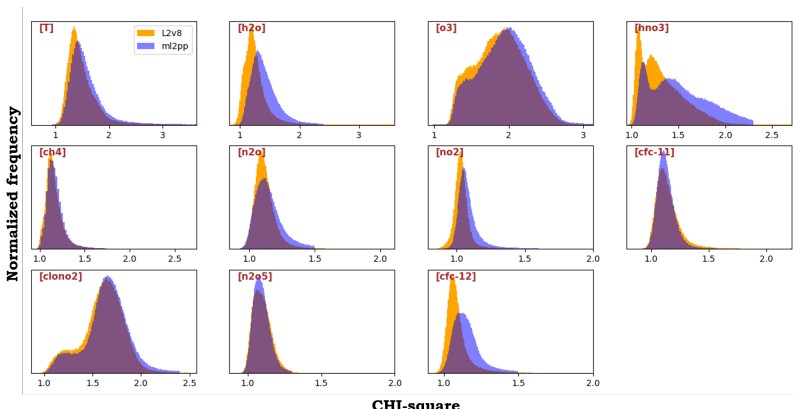

**Figure 1.** Chi-square distribution of all FR NOM L2 V8 (orange) and L2 V7 (blue) profiles for some representative trace species.

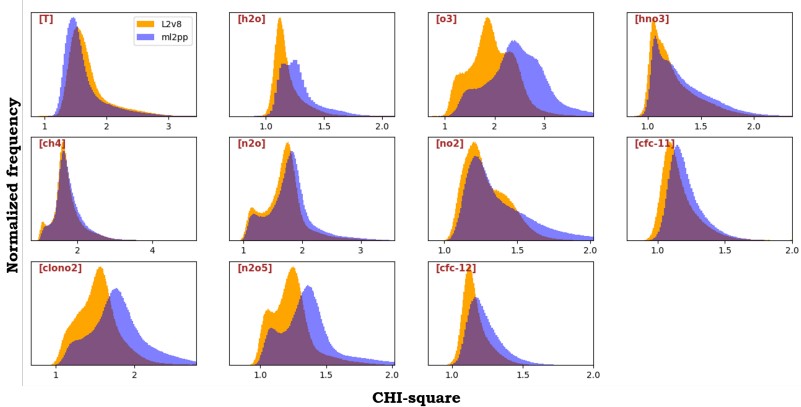

**Figure 2.** Chi-square distribution of all OR NOM L2 V8 (orange) and L2 V7 (blue) profiles for some representative trace species.

are mainly due to the modifications in the building of the profiles of the gases that are spectrally interfering with the retrieved gas, and that are retrieved, in a limited vertical range, in a previous step of the retrieval chain. The improved spectroscopic data used in V8 and the introduced horizontal gradient model also contribute, albeit with a lesser extent, to the observed decrease in chi-square.

## 4.1   Model of the horizontal variability

Systematic differences between profiles retrieved from the measurements acquired in the day and the night parts of the satellite orbit were observed, for species for which a diurnal variability is not expected, in the L2 products V7 and earlier. As described in Sect. 2, with the only exception of high latitude measurements for which the illumination depends on the season, in the MIPAS data set day / night differences are a synonym for descending / ascending differences.



Systematic day / night differences were first noticed by Kiefer et al. (2010) who compared zonal averages of MIPAS mea-
surements owing to the ascending and to the descending parts of the satellite orbit. The observed differences are of the order
of the retrieval noise error, therefore they are only visible when averages of large data sets are compared. Kiefer et al. (2010)
attributed this effect to the missing model of the horizontal variability of the atmosphere within the radiative transfer code. In
fact, assuming a linear variation of the atmospheric state parameters with latitude, a given North-South gradient has opposite-
in-sign projections along the instrument line of sight, depending on whether the measurement is acquired in the ascending or
in the descending part of the orbit. As a consequence, a forward model assuming a horizontally homogeneous atmosphere will
simulate radiances affected by opposite-in-sign systematic errors in the ascending and in the descending parts of the orbit. In
turn, this forward model error will map into systematic overestimates / underestimates of the retrieved profiles in the ascending
/ descending sections of the orbit. Ascending-descending differences were shown (Kiefer et al., 2010) to vanish when adopting
a full tomographic approach (Carlotti et al., 2001), confirming the need for taking into account horizontal gradients. Extensive
tests (Kiefer et al., 2010; Castelli et al., 2016) showed that modelling the horizontal variability of the atmosphere with an exter-
nally supplied horizontal gradient (HG) could reduce significantly the systematic ascending-descending differences, provided
that a sufficiently accurate gradient estimate is used. According to these results the ORM was modified so that, starting from
V8, it models an height-dependent HG of both temperature and gases VMR. HGs are not retrieved, they are computed from
external profile databases. Allowed profile data sources are the ECMWF database, results of previous L2 MIPAS/ESA data
processing and the IG2 database (see Sect. 4.3) which also includes the climatological variability with latitude. Sensitivity tests
have shown that the HGs determined on the basis of the climatological latitude variability tabulated in the IG2 database, locally,
are not sufficiently accurate. HGs estimated from earlier MIPAS reprocessings are generally accurate, however, sporadically,
they may show unphysical oscillations. Since HGs determined from the ECMWF database proved to be sufficiently accurate
and reliable, in the ORM V8 reprocessing we used these HGs for Temperature, $H_2O$ and $O_3$, and set to zero the HGs of the
other gases. The HGs are calculated as difference quotients at fixed altitudes along the orbit plane. For each scan, we apply the
gradient to the atmosphere up to a distance $\delta OC$ from the tangent point, where OC (orbital coordinate) is the polar coordinate
in the orbit plane. We set $\delta OC$ as the average inter-scan distance (about $4°$). Since the effective angular distance between scans
differs from scan to scan, corrective actions have been taken to avoid unphysical extrapolations of the atmospheric fields in the
OC domain.

Removing the assumption of horizontal homogeneity of the atmosphere implies that the Snell's law could not be exploited
any longer, as in the earlier ORM versions. As a consequence, a new ray-tracing algorithm has been developed in order to
calculate the Curtis-Godson integrals, defining the equivalent quantities of each layer. The new algorithm (Ridolfi and Sgheri,
2014) solves the Eikonal equation in Cartesian coordinates using a multi-step predictor-corrector method, with an adaptive step
that depends on the curvature of the ray-paths.

Temperature and CFC-11 were identified by Kiefer et al. (2010) as the most critical target parameters, showing the largest
ascending / descending differences. Figures 3 and 4 show the average vertical profiles of ascending / descending differences for
Temperature and CFC-11, respectively. The averaging period covers the measurements acquired in the months of December
of the years from 2006 to 2011 (i.e. the OR part of the mission). The differences were binned into ten latitude intervals.



We see that, in general, for temperature and CFC-11 the ascending / descending differences in V8 products (red curves)
are significantly smaller than in V7 products (blue curves). Specifically, the introduced HG model reduces the temperature
systematic differences (in about 1 to 2 K) at mid- to higher latitudes, while preserving the real ascending / descending (night /
day) differences at the tropics, due to solar tides.

The systematic ascending / descending differences are linked to the meridional variability of the atmosphere, therefore, as
expected, they depend on season. The seasonality of these differences is illustrated in Figs 5 and 6. For a selected latitude band
$(45° – 60°S)$, these figures show the time series of the monthly average temperature (Fig. 5) and CFC-11 (Fig. 6) ascending
/ descending difference profiles for V7 (left map) and V8 (right map). Although still visible, the seasonality of the ascending
/ descending differences is much less pronounced in V8 data as compared to V7. The remaining seasonality in V8 could be
due to the fact that our model of the horizontal variability of the atmosphere is a first order model, i.e. we only model a linear
variation using a horizontal gradient.

## 4.2  Updated spectroscopic data

Full-physics modelling of the measurements requires the knowledge of the spectroscopic parameters of the trace species emit-
ting in the spectral region to be simulated. Indeed, errors in the spectroscopic parameters are estimated to provide one of the
major contributions to the total error budget of the retrieved profiles (Dudhia, 2021). The crucial role played by the quality
of spectroscopy in the quality of retrieved profiles has motivated the development of a dedicated spectroscopic database for
MIPAS with an activity that proceeded in parallel with the development of the L2 processor. For the analysis with ORM V8, the
MIPAS dedicated spectroscopic database HITRAN_mipas_pf4.45 was used which is the evolution of HITRAN_mipas_pf3_2
spectroscopic database used for the processing of V6 and V7 L2 data set. The format of the spectroscopic database pf4.45 is
compliant with HITRAN 2004.

HITRAN_mipas_pf4.45 is based on HITRAN08 (Rothman et al., 2009), but spectroscopic parameters for the molecules
$O_2$, $SO_2$, OCS, $CH_3Cl$, $C_2H_2$, $C_2H_6$ are taken from HITRAN 2012 (Rothman et al., 2012). The spectroscopic parameters of
$HNO_3$ were derived by Perrin et al. (2016), the spectroscopic data for the $COCl_2$ were derived by Tchana et al. (2015). Both
$HNO_3$ and $COCl_2$ data are now contained in HITRAN 2016 (Gordon et al., 2017). Spectroscopic data for the new molecule
$C_3H_8$ (Flaud et al., 2010; Nixon et al., 2009), which are not present in HITRAN data set up to 2016, have been included in the
pf4.45 data set. Among the species for which spectroscopic data have changed significantly with respect to previous MIPAS
spectroscopic database HITRAN_mipas_pf3_2 we have to mention HCN (see Sect. 4.2.2). Spectroscopic line data relative to
HOBr are still excluded from the database as the available data are for pure rotational transitions and are outside MIPAS bands.
Also line data relative to $CF_4$ are still excluded from the database since their quality is very poor.

For molecules which exhibit very dense line-by-line spectra that are extremely difficult to model, or for which the indi-
vidual transitions have not been assigned or are not accurate enough or are of poor quality, cross-sections measured in the
laboratory for atmospheric pressure and temperature ranges are used. It is worth noticing that cross-sections also have the ad-
vantage of incorporating various spectroscopic effects, such as line coupling and pressure shifts. Measured cross-sections are
used for the following molecules: CFC$-$11, CFC$-$12, CFC$-$13, CFC$-$14, HCFC$-$21, HCFC$-$22, CFC$-$113, CFC$-$114,

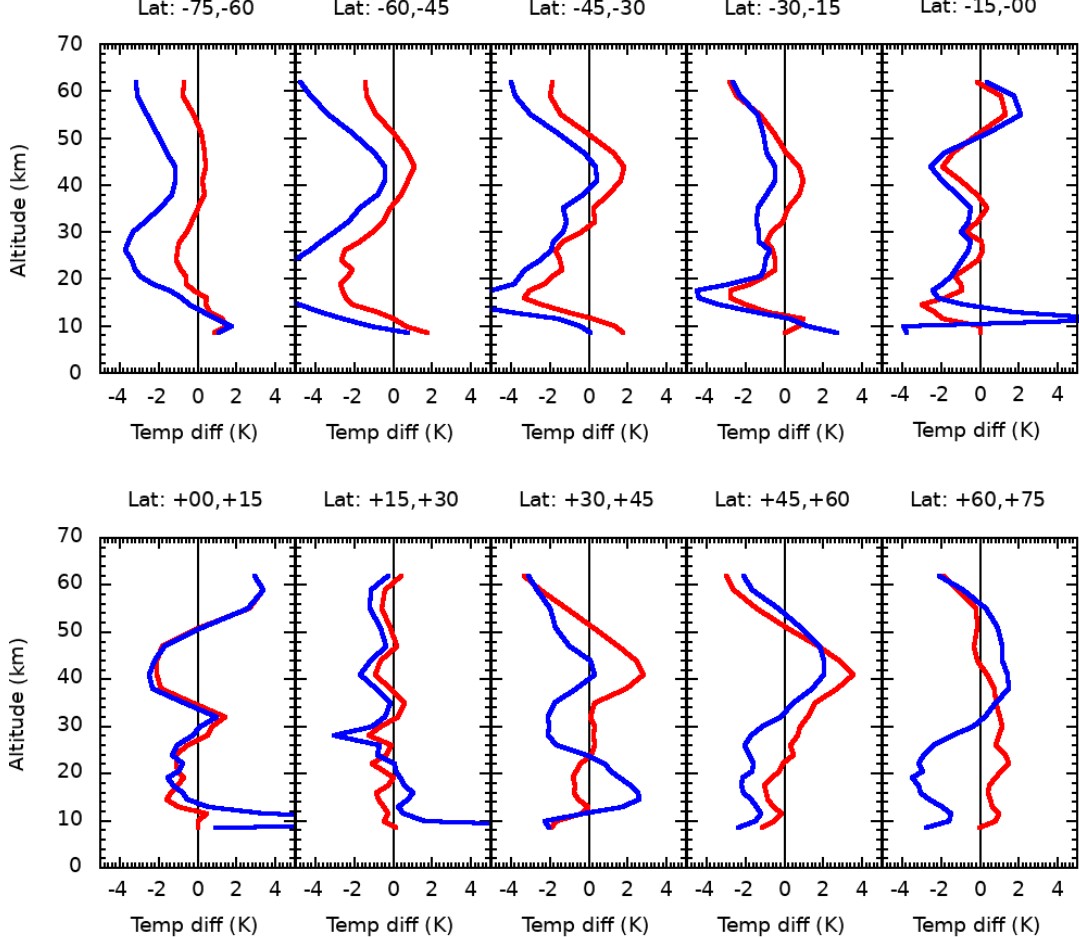

**Figure 3.** Temperature ascending-descending average difference profiles for V7 (blue) and V8 (red) MIPAS/ESA products. Averaging extends to the measurements acquired in the month of December of the years from 2006 to 2011 (OR part of the mission). The different panels refer to different latitude bands as indicated on the top of the panels.

CFC$-115$, $CCl_4$, $ClONO_2$, $N_2O_5$, $HNO_4$ and $SF_6$. Among these, new cross-sections have been used in ORM V8 for CFC$-12$, CFC$-14$, HCFC$-22$, $CCl_4$, $ClONO_2$, $HNO_4$, taken from HITRAN 2016 (Rothman et al., 2017), CFC$-11$ (Har-

rison, 2018), CFC$-113$ (Le Bris et al., 2011) and $SF_6$ (Driddi et al., 2021).

It is important to note that MIPAS measurements themselves were used for some molecules to verify that the new spectroscopic parameters obtained by laboratory measurements allowed to reduce the differences between the observations and the simulations and, thanks to the broadband spectra of MIPAS, to check the consistency of the line parameters of a given molecule in its different absorption bands. This is the case for $HNO_3$, for which an absolute intensity calibration was performed to "con-

vert" the relative line intensities at 7.6 $\mu$m to absolute intensities. This was done by comparing $HNO_3$ VMR retrieved profiles



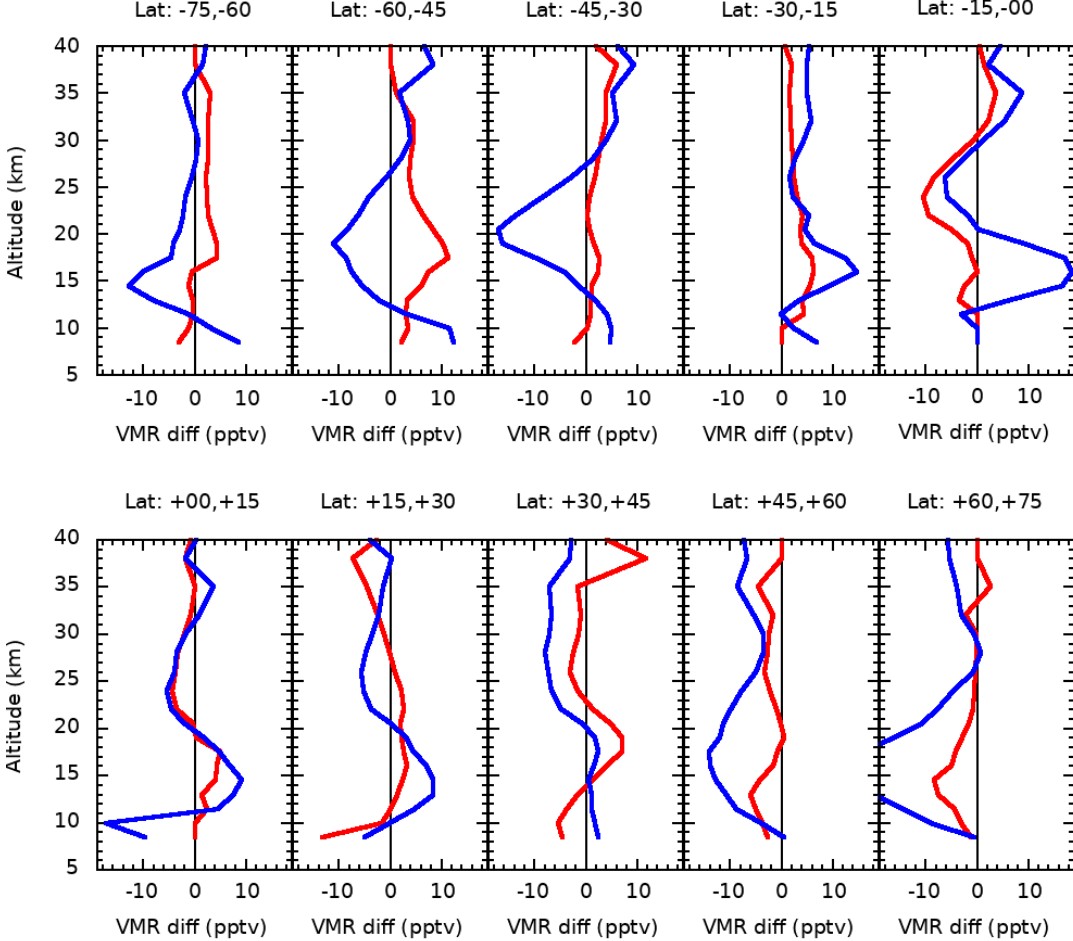

**Figure 4.** CFC-11 ascending-descending average difference profiles for V7 (blue) and V8 (red) MIPAS/ESA products. Averaging extends to the measurements acquired in the month of December of the years from 2006 to 2011 (OR part of the mission). The different panels refer to different latitude bands as indicated on the top of the panels.

from MIPAS radiances in either the 7.6 or 11 $\mu$m regions. A multiplicative factor was applied to all the line intensities at 7.6 $\mu$m so that in the height range of the $HNO_3$ VMR peak (21–24 km), the VMR retrieved using the 7.6 $\mu$m interval matched the one retrieved using the 11 $\mu$m region, leading to better consistency between the 11 and 7.6 $\mu$m regions (Perrin et al., 2016).

In Figs 7, 8 and 9 three examples are shown of spectral intervals selected for the retrieval of $HNO_3$, $COF_2$ and $CFC-12$ 265 VMR, where differences in the residuals come from the changed spectroscopic data only. It is evident that the use of the new spectroscopic database and the new cross-sections significantly reduces the residuals.





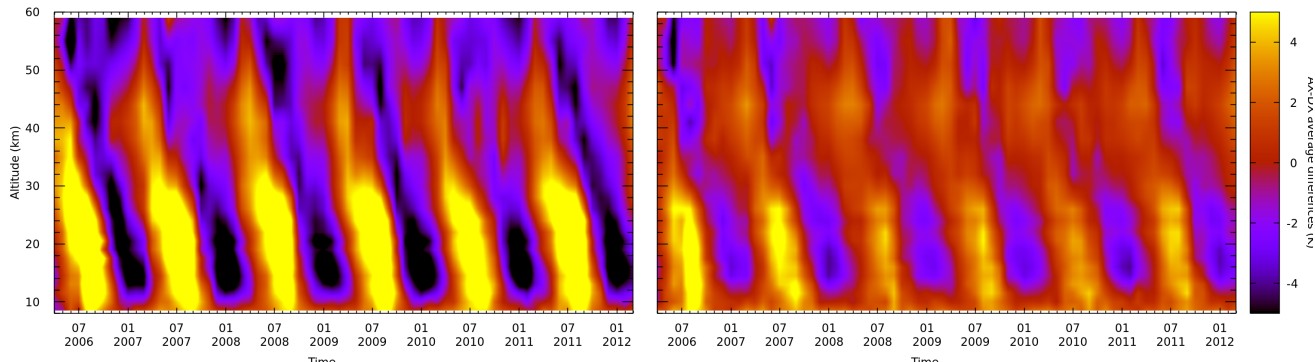

**Figure 5.** Time series of monthly average temperature ascending-descending difference profiles for v7 (left) and v8.22 (right) MIPAS/ESA products. The maps refer to the latitude band from 45S to 60S.

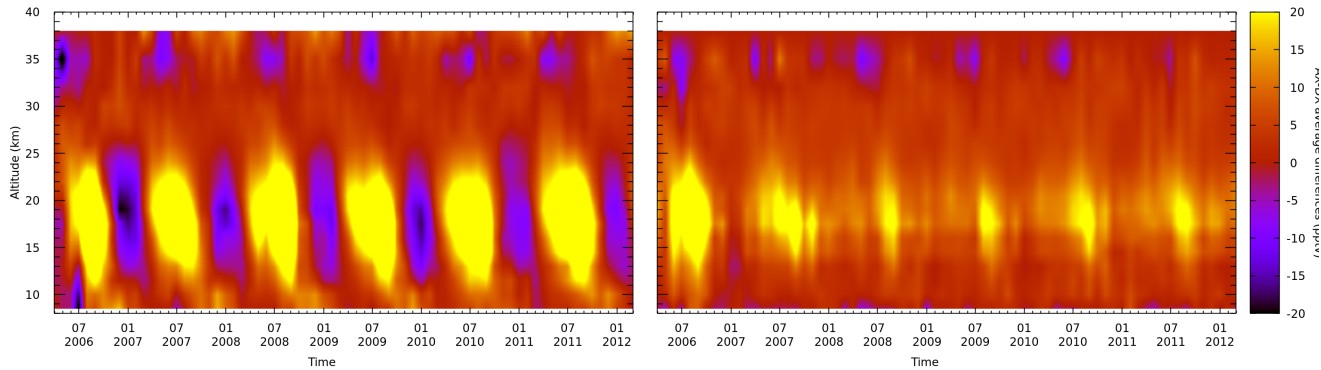

**Figure 6.** Time series of monthly average CFC-11 ascending-descending difference profiles for v7 (left) and v8.22 (right) MIPAS/ESA products. The maps refer to the latitude band from 45S to 60S.

### 4.2.1 Impact on retrieved species

We have shown some examples of how the use of the new spectroscopic parameters and cross-sections improves the simulations of the observations, but they lead also to significant changes in the retrieved profiles. The validation of the retrieved profiles is outside the scope of this paper. Here we only describe the impact of the changes in the spectroscopic line data and the cross-sections on the retrieved profiles for the most affected trace species.

Spectroscopic line data updates affect mostly $HNO_3$, HCN and $COCl_2$. The use of the updated spectroscopic database for $HNO_3$ causes systematically larger profiles between 100 and 10 hPa, with the largest differences being 0.7 (0.2) ppbv for the FR (OR) measurements around the peak of the profile in the antarctic region (see Fig. 10), corresponding to differences of about +5% (2.5%) for the FR (OR) measurements.



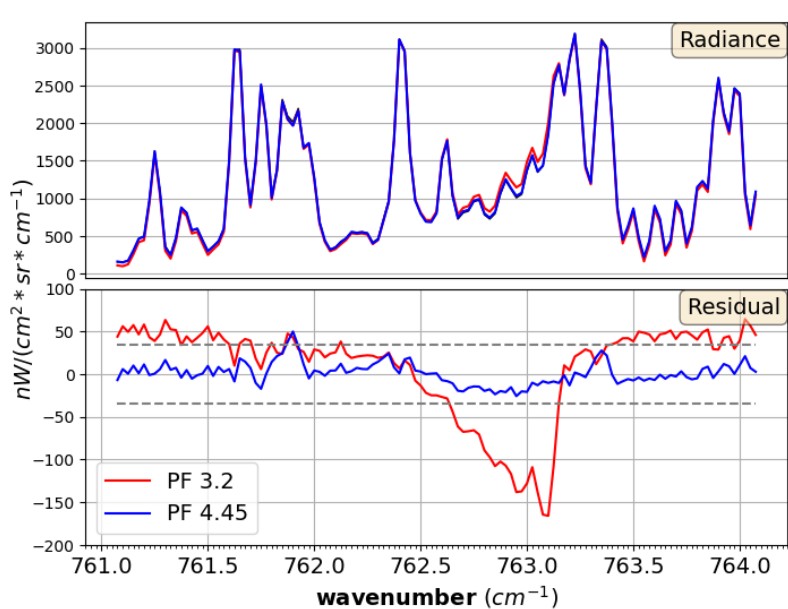

**Figure 7.** Upper plot: Observed spectrum (black curve, under the blue one) and simulations at a limb tangent height of 24 km with the old (red curve) and the new (blue curve) spectroscopic parameters for a microwindow used for $HNO_3$ retrieval for FR measurements. Lower plot: Residuals obtained with simulations generated with the old spectroscopic database (red curve) and the new one (blue curve), compared with the measurement noise (grey curves).

For HCN the changed spectroscopy is responsible of an even larger difference, greater than 20 pptv between 200 and 10 hPa (see Fig. 11) corresponding to 15 - 20%. The reason is that a major update has been accomplished for the hydrogen cyanide line list since HITRAN 04. Line positions and intensities throughout the infrared have been revisited by Maki et al. (1996, 2000). The improvements apply to the three isotopologues present in HITRAN in the pure-rotation region and in the infrared
from 500 to 3425 $cm^{-1}$. The new intensities are about 1.16 times larger than the previous ones.

For the impact of the changes in $COCl_2$ spectroscopy on the retrieved profiles please refer to Pettinari et al. (2021).

Systematic differences in the retrieved trace species attributable to changes in the used cross-sections are found in $CCl_4$, $CFC-11$, $CFC-12$ and $HCFC-22$ retrieved profiles. In general, for all the four trace species the new cross-sections are characterised by a higher spectral resolution and better signal to noise ratio at low pressure than the previous ones.
The use of the new cross-sections for $HCFC-22$ (Harrison, 2016) leads, above 200 hPa, to retrieved $HCFC-22$ VMRs between 10 and 25 pptv smaller than with the old cross-sections (Varanasi, private communication; Clerbaux et al., 1993) (see Fig. 12), corresponding to differences of about 10%. A possible explanation for this difference is that the Q branches are reasonably sharp, especially near 829 $cm^{-1}$, and very sensitive to pressure. So even though the overall integrated band strength is quite the same, differences can be due to the extended pressure coverage of the new cross-sections.

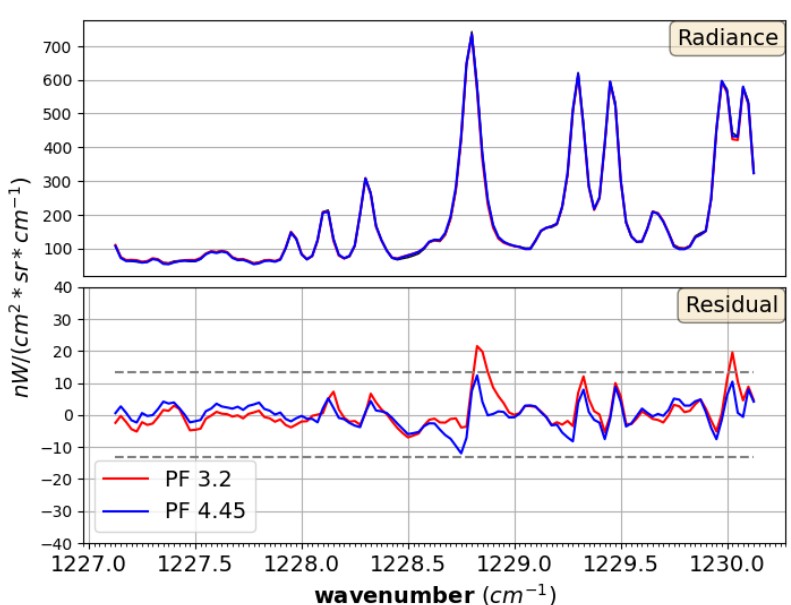

**Figure 8.** Upper plot: Observed spectrum (black curve, under the blue one) and simulations at $18\,\mathrm{km}$ with the old (red curve) and the new (blue curve) spectroscopic parameters for a microwindow used for $COF_2$ retrieval for FR measurements. Lower plot: Residuals obtained with simulations generated with the old spectroscopic database (red curve) and the new one (blue curve), compared with the measurement noise (grey curves)

Concerning the other retrieved profiles, V8 $CFC-11$ VMRs obtained with the new cross-sections (Harrison, 2018) are up to 20-30 pptv smaller than with the old ones (Varanasi, private communication) (see Fig. 13), corresponding to a percent difference of up to 5%. Differences in $CCl_4$ VMRs retrieved with the new (Harrison et al., 2016) and the old (Varanasi, private communication) cross-sections vary between 0 at $100\,\mathrm{hPa}$ for the tropical bands to -5 to -10 $\mathrm{pptv}$ from 200 to 20 $\mathrm{hPa}$ for the other latitude bands and altitudes (see Fig. 14). Finally, the new $CFC-12$ cross-sections (Harrison, 2015) are responsible for an increase in the retrieved $CFC-12$ profiles with respect to the old cross-sections (Varanasi, private communication) which is maximum at the lowest altitudes and equals about 50 pptv , then the difference gradually reduces with altitude, becoming negligible at $100\,\mathrm{hPa}$, and slightly negative above (Fig. 15) .

For $CFC-12$, it is interesting to compare the differences in the retrieved profiles between V8 and V6 products with the results of the validation of V6 data with the balloon BONBON measurements (Engel et al., 2016), that uses gas chromatography, for an indirect verification on the improvements implemented in the V8 processor. The use of the new cross-sections, combined with the other changes implemented in the code, produces differences between V8 and V6 of opposite sign in the two phases of the mission (positive in the OR, negative in the FR) (Fig. 16), which compensates the bias between MIPAS V6 and BONBON measurements (negative in the OR, positive in the FR, see Fig. 13 of Engel et al. (2016)) leading to a better agreement.



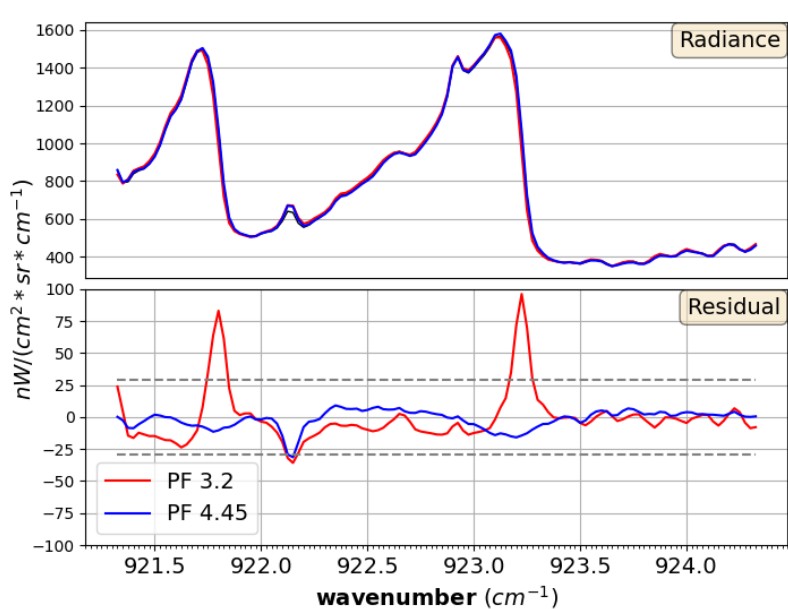

**Figure 9.** Upper plot: Observed spectrum (black curve, under blue one) and simulations at 15 km with the old and the new spectroscopic parameters for a microwindow used for CFC−12 retrieval for FR measurements. Lower plot: Residuals obtained with simulations generated with the old spectroscopic database (red curve) and the new one (blue curve), compared with the measurement noise (grey curves)

For the other species validated with the same technique, namely $CH_4$, $N_2O$ and CFC−11 (Engel et al., 2016), the detected
biases of V6 products wrt BONBON (up to 300 ppbv at 15 km for FR measurements in $CH_4$, up to 50 ppbv at 15 km for FR measurements in $N_2O$ and up to 40 pptv for CFC−11 in both phases of the mission) were already partially reduced in V7 products thanks to the use of new $N_2O$ and $CH_4$ microwindows for the analysis of FR measurements and to the handling of the interference of $COCl_2$ in CFC−11 retrievals (De Laurentis et al., 2016). Furthermore, no significant differences are found in V8 $CH_4$ and $N_2O$ wrt corresponding V7 products and the effect of the new cross-sections for CFC−11 is smaller than the
effect of the unaccounted $COCl_2$ interference in V6 CFC−11 profiles.

### 4.3 Use of the a priori knowledge of the atmosphere

In order to perform the retrieval, it is necessary to define the initial and a-priori state of the atmosphere. The atmosphere is described by the vertical distributions of pressure and temperature, as well as of the volume mixing ratio (VMR) of both the retrieval targets and the interfering species, and, with the handling of horizontal inhomogeneities, also of the horizontal
(latitudinal) gradients of the considered species. The choice of these profiles, in particular of the ones which are not target of the retrieval, is important, because a wrong assumption may introduce systematic errors in the retrieved quantities. In general, the following databases have been developed for defining the state of the atmosphere:



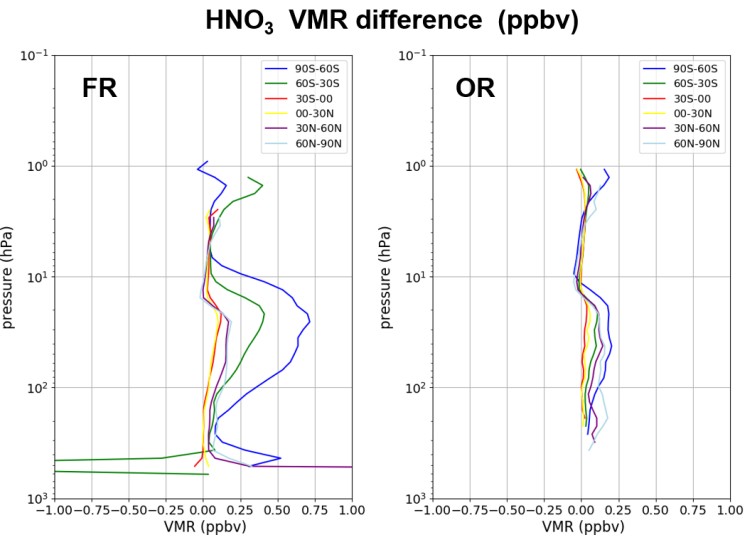

**Figure 10.** Absolute difference between the mean of the $HNO_3$ profiles retrieved using the new (mipas_pf4.45) and the old (mipas_pf3.2) spectroscopic databases at different latitude bands for selected orbits in FR phase (left plot) and OR phase (right plot).

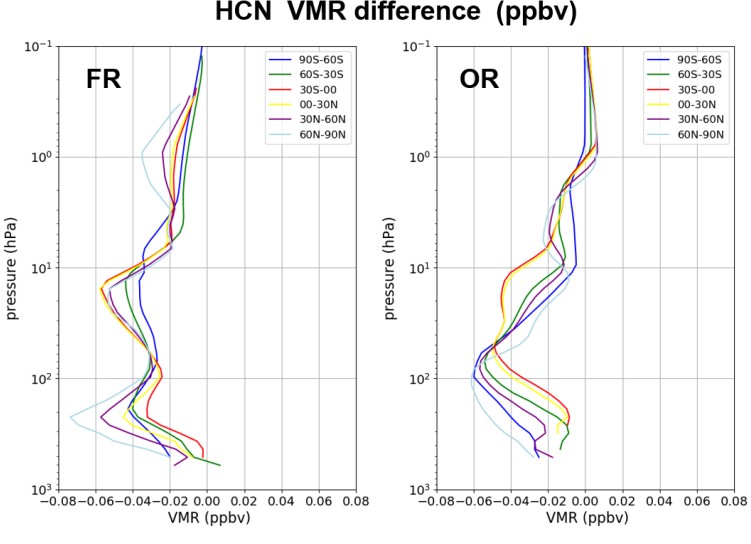

**Figure 11.** Same as Fig. 10, but for the HCN profiles.





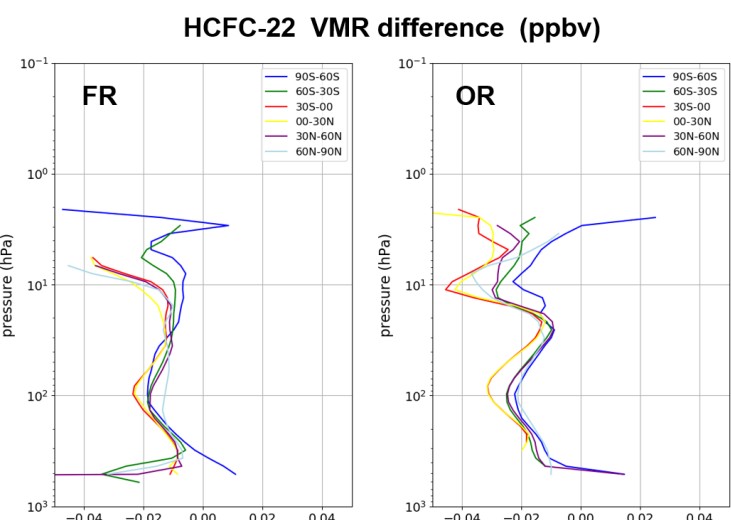

**Figure 12.** Same as Fig. 10, but for the HCFC−22 profiles.

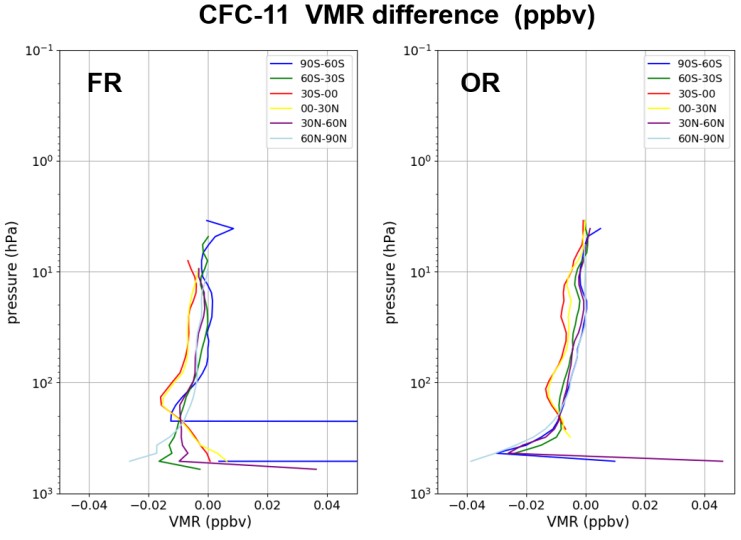

**Figure 13.** Same as Fig. 10, but for the CFC−11 profiles.





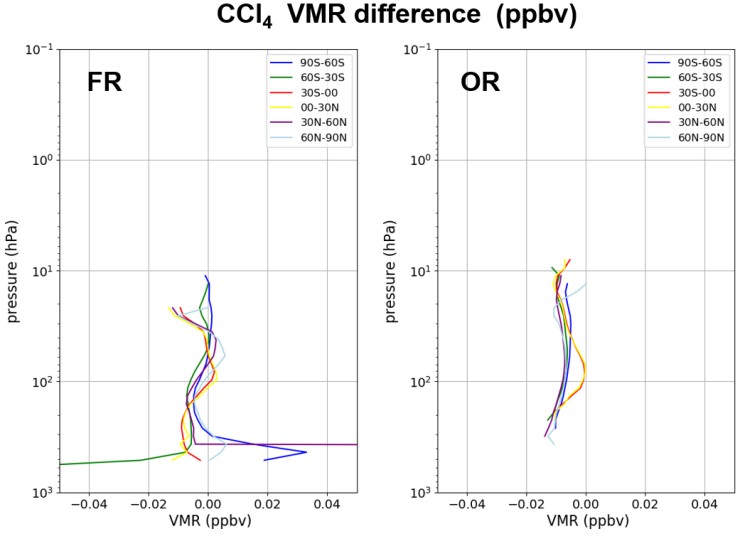

**Figure 14.** Difference in the retrieved $CCl_4$ profile with the new and the old spectroscopic database for selected orbits in FR phase (left plot) and OR phase (right plot).

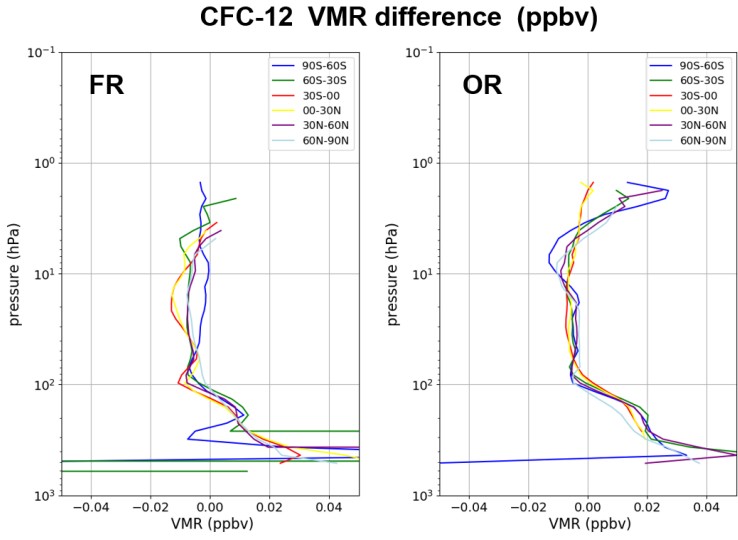

**Figure 15.** Same as Fig. 10, but for the $CFC-12$ profiles.

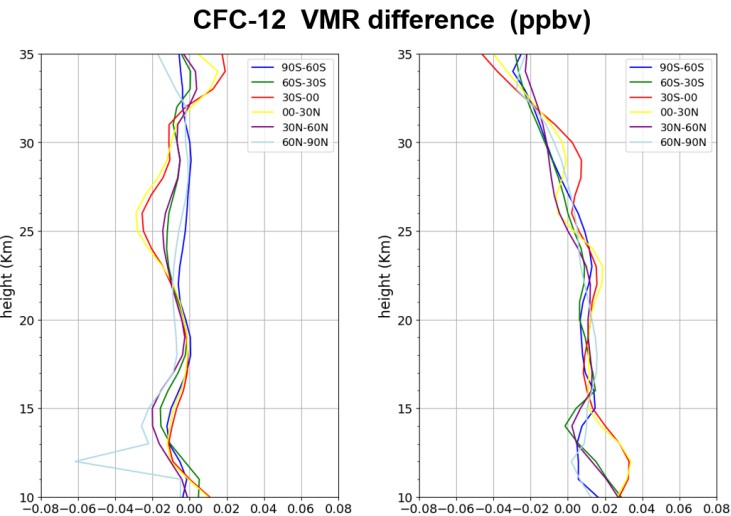

**Figure 16.** Absolute difference between MIPAS V8 and MIPAS V6 CFC−12 profiles at different latitude bands for selected orbits in FR (left plot) and OR (right plot) phases.

- The Level 2 Initial Guess (IG2) data set, consisting of a set of climatological profiles of the global atmosphere for six latitude bands, four seasons (January, April, July and October), and for all years of the MIPAS mission in both nighttime and daytime conditions. These profiles are available for all targets and interfering species in the altitude range 0–120 km. The methodology used for generating these data is based on Remedios et al. (2007) and the improvements implemented in version V5.4 (used for the L2 V8 reprocessing) are described in Sect. 4.3.1.

- Profiles from ECMWF ERA-Interim reanalysis, that have been chosen for each scan as the closest to MIPAS measurements: this data set includes pressure, temperature, water vapour and ozone profiles in the altitude range 0–65 km. The way they are selected is described in Sect. 4.3.2.

- For the target species, retrieved in sequence from each measured scan, the following profiles can also be used, if available and of 'good' quality (see Sect.6.2): profiles from previous retrievals of the current measurement scan; retrieved profiles from the previous scan; retrieved profiles from previous MIPAS reprocessings.

Among the different databases, only the IG2 profiles are reported on the altitude range from 0 to 120 km. The other profiles are extended on the same vertical range by retaining the shape of the corresponding IG2 profile, and scaling it with the ratio between the value of the profile at the extremes of its altitude range and the corresponding climatological value interpolated at the same pressure.

The code uses a priority system to determine the best possible choice for the target species and the interfering species.



### 4.3.1 Initial Guess 2 data set

The IG2 database consists of the climatological profiles for the following six latitude bands: 90ºN to 65ºN, 65ºN to 20ºN, 20ºN to 0, 0 to 20ºS, 20ºS to 65ºS and 65ºS to 90ºS for January, April, July and October. Profiles are represented on a 1 km step vertical grid from 0 km to 120 km for pressure (mb), temperature (K) and volume mixing ratios (ppmv) of the following species: $N_2$, $O_2$, $C_2H_2$, $C_2H_6$, $C_3H_8$, $CO_2$, $O_3$, $H_2O$, $CH_4$, $N_2O$, CFC−11, CFC−12, CFC−13, CFC−14, CFC−21, HCFC−22, CFC−113, CFC−114, CFC−115, $CH_3Cl$, $CCl_4$, HCN, $NH_3$, $SF_6$, $HNO_3$, $HNO_4$, NO, $NO_2$, $SO_2$, CO, HOCl,

ClO, $H_2O_2$, $N_2O_5$, OCS, $ClONO_2$, $COF_2$, $COCl_2$, HDO and PAN. Daytime and nighttime profiles are provided for the trace gas species that display strong diurnal signatures, namely CH4, ClO, $ClONO_2$, $H_2O$, $HNO_3$, $HNO_4$, HOCl, $N_2O_5$, $N_2O$, $NO_2$ and $O_3$. For specific gases (i.e. chlorofluorocarbons and $CO_2$) concentrations change annually.

The creation of the IG2 profiles follows the methodology of Remedios et al. (2007) in which several data sources are selected for specific regions of the atmosphere and merged to create a full vertical concentration profile from the ground to

120 km altitude. The following data sets are used for the creation of the V5 IG2 profiles: URAP (UARS Reference atmosphere project) profiles (Remedios, 1998), ACE-FTS on SCISAT v3.6 data set (Bernath et al., 2003), SLIMCAT model profiles (see description below), GEOS-Chem model profiles (Bey et al., 2001; Shephard, private communication), IAA profiles derived for upper altitudes and diurnal variations as required for non-LTE calculations (see description below), mean profiles calculated from ensembles of MIPAS-Oxford profiles (2008–2009) for the MIPAS target species (Dudhia, 2008). A summary of the input

data used for IG2 V5 profiles is provided in Table 1. CFCs and organics are constrained by the V3 ACE-FTS data set with further filtering to remove negative volume mixing ratios. Variable smoothing lengths, depending on species and atmospheric lifetime, are required to smooth out small-scale vertical variability.

For the generation of a day and night data set, the creation of V5 differs only so as to incorporate the diurnal differences expected for some species. This is done by changing some of the input data sets to accommodate other data sources that

possess day and night variations. In particular, in the lowest altitude range diurnal variation from either the University of Leeds SLIMCAT model (see below) or the climatology from MIPAS products were implemented into the V5 profile creation procedure, while the IAA data set (see below) was mainly used above this altitude range . The output results in two profiles created per gas (per season and per latitude band) rather than a single mean profile.

The University of Leeds SLIMCAT model is a 3D stratospheric chemical transport model that calculates atmospheric

abundances of a number of chemical species (Chipperfield, 2006). In a combination with TOMCAT, the inclusive TOM-CAT/SLIMCAT model calculates the global chemical tracer fields from the surface to 80 km for short-lived chemical species, steady state and source gases, using meteorological fields from ECMWF. Here, for the stratosphere, vertical transport is calculated by the CCMRAD (Briegleb, 1992) radiation scheme that encompasses the longwave and shortwave radiation domains (from the surface to 0 hPa). Advection of chemical tracers is performed by the Prather scheme (Prather, 1986). A diurnally

varying SLIMCAT data set was computed specifically for the creation of IG2 diurnally varying profiles. Chemical calculations were performed on a monthly scale from pre-Pinatubo years to years including ENVISAT coverage (1990 to 2008), driven by ECMWF reanalysis, and calculated on a 1 km vertical grid extending from the surface to 60 km (1999 – 2005) and 80 km





(2006 – 2008). The chemical tracers included in these calculations are $O_3$, $O$, $O(^1D)$, $N$, $NO$, $NO_2$, $NO_3$, $N_2O_5$, $HNO_3$,
$ClONO_2$, $ClO$, $HCl$, $HNO_4$, $HOCl$, $Cl$, $Cl_2O_2$, $OClO$, $Br$, $BrO$, $BrONO_2$, $BrCl$, $HBr$, $H_2O_2$, $HOBr$, $CO$, $CH_4$, $N_2O$, $H$,
$OH$, $HO_2$, $H_2O$, $H_2SO_4$, $CFC-11$, $CFC-12$, $CH_3Br$, $CH_2O$, $HF$, $COF_2$ and $COFCl$. Data are provided as zonally aver-
aged day and night concentration profiles, zonal minimum/maximum day and night profiles and standard deviation day and
night profiles. Data are gridded onto a 5° latitude grid ranging from –85° to 85° and on a 1 km vertical grid.

   Climatology from MIPAS products was generated for a check on diurnal variation provided by SLIMCAT. In particular
MIPAS-Oxford profiles (Dudhia, 2008) were used. A Z-Test filter (Daszykowski et al., 2007) was applied to the MIPAS-
Oxford data to remove any outliers. Some control tests performed with ORM, consisting in pressure and temperature retrievals
for selected orbits covering the four seasons, performed using as the interfering species profiles the climatological profiles
taken from either the V4.1 IG2 database or the new diurnally varying IG2 database, revealed that larger chi-square values were
found compared to V4.1 IG2 database when using $N_2O_5$ profiles derived by SLIMCAT, with the largest discrepancies being in
the polar regions. It was found that the $N_2O_5$ profiles generated by SLIMCAT did not appear to suitably represent the diurnal
variation (10 am / 10 pm) expected for these species, especially in the polar regions, despite good apparent agreement with the
twilight occultation observations of ACE-FTS. At the end mean profiles calculated from ensembles of MIPAS-Oxford $N_2O_5$
profiles (2008–2009) were preferred to SLIMCAT profiles to define the diurnal information for generating the V5 IG2 profiles.
To keep consistency between all of the diurnal species retrieved by MIPAS, similar V5 IG2 profiles were generated for all
operational species ($O_3$, $N_2O$, $ClONO_2$, $H_2O$, $CH_4$, $HNO_3$ and $NO_2$).
Above 80 km, the IAA data set is mainly used. This supplies a set of climatological profiles extending from the surface up
to 200 km for the same six latitude bands, four seasons and nighttime and daytime conditions of the IG2 database. For the
generation of these profiles, the SLIMCAT profiles available up to 80 km were extended and/or modified up to 200 km as
detailed in Table A1. The data set contains profiles for: a) pressure and temperature, which were taken from the IG database
version V4 (Remedios et al., 2007) below 70 km and from the MSIS model (Picone et al., 2002) above that altitude; b) vmr
profiles for $H_2O$, $CO_2$, $O_3$, $N_2O$, $CO$, $CH_4$, $NO$, $NO_2$, $HNO_3$, $OH$, $ClO$, $HO_2$, $O$, $O(^1D)$, $N$, $H$, $N_2$, $O_2$, and $HCN$; and c)
non-LTE relevant parameters as the photo-dissociation rates of $O_2$ and $NO$, $J_{O2}$ and $J_{NO}$, respectively.

   For generating this data set we used a simple 1D chemistry model (the IAA box model), MSIS model (Picone et al., 2002),
the 2D model of Garcia and Solomon (1994), and the NOEM (Marsh et al., 2004) for NO. Concerning the diurnal variations, the
calculations for particular reference days were performed (see Table 2 in López-Puertas et al. (2009)) within the corresponding
season (i.e. March-May for April) at 10 am/pm local time. These reference days were chosen such that Solar Zenith Angle
(SZA) at 10 am/pm reflects the average SZA of all MIPAS observations in the corresponding latitudinal and seasonal band
at day (SZA<90) and night (SZA>90) conditions. The photo-dissociation coefficients, $J$'s, were calculated in the IAA box
model with the TUV model (Madronich and Flocke, 1998), except for $J_{NO}$ that was taken from the Minschwaner and Siskind
(1993) parameterization. More details on the generation of this data set can be found in López-Puertas et al. (2009) and in
Sec. 3.1 of Funke et al. (2012).





| IG2 Species | Data Source | IG2 Species | Data Source |
|---|---|---|---|
| $C_2H_2$ | ACE-FTS v3.6 | CFC$-$115 | URAP |
| $C_2H_6$ | ACE-FTS v3.6 | $H_2O$ | MIPAS-Oxford, IAA |
| $C_3H_8$ | ACE-FTS v3.6 | $H_2O_2$ | URAP, SA |
| $CCl_4$ | ACE-FTS v3.6 | HCN | ACE-FTS v3.6 |
| $CH_3Cl$ | URAP | HDO | ACE-FTS v3.6 |
| $CH_4$ | MIPAS-Oxford, IAA | $HNO_3$ | MIPAS-Oxford, IAA |
| ClO | URAP, IAA | $HNO_4$ | SLIMCAT, IAA |
| $ClONO_2$ | MIPAS-Oxford, IAA | HOCl | SLIMCAT, IAA |
| CO | SLIMCAT, IAA | $N_2$ | URAP |
| $CO_2$ | URAP, IAA | $N_2O$ | MIPAS-Oxford, IAA |
| $COCl_2$ | ACE-FTS v3.6 | $N_2O_5$ | SLIMCAT, IAA |
| $COF_2$ | ACE-FTS v3.6 | NH3 | GEOS-Chem |
| CFC$-$11 | ACE-FTS v3.6 | NO | URAP, IAA |
| CFC$-$12 | ACE-FTS v3.6 | $NO_2$ | MIPAS-Oxford, IAA |
| CFC$-$13 | URAP | $O_2$ | URAP |
| CFC$-$14 | ACE-FTS v3.6, URAP | $O_3$ | MIPAS-Oxford, IAA |
| CFC$-$21 | URAP | OCS | ACS-FTS v3.6 |
| HCFC$-$22 | ACE-FTS v3.6, URAP | PAN | ACS-FTS v3.6, SLIMCAT |
| CFC$-$113 | ACE-FTS v3.6, URAP | $SF_6$ | ACS-FTS v3.6, URAP |
| CFC$-$114 | URAP | $SO_2$ | URAP, SA |

**Table 1.** Summary of input data for IG2 V5 profiles. For previous versions we refer to Remedios et al. (2007). The standard atmospheres (SA) database is a set of climatological profiles that represent the global atmosphere under conditions of varying atmospheric state (Remedios et al., 2007). Other data sets used are described in the text.

### 4.3.2 ECMWF data set

The so-called ECMWF data set is taken from ECMWF ERA-Interim (Dee et al., 2011), the latest global atmospheric reanalysis of ECMWF model results available at the start of MIPAS data reprocessing. The data set covers the period 1979 to 2019 with a spatial resolution of approximately 79 km (T255 spectral) on 60 vertical levels from the surface up to 0.1 hPa. ERA-Interim
data include temperature, humidity and ozone profiles, which are made available as 6-hourly atmospheric fields.

Data on model levels were adopted instead of pressure levels in order to obtain a greater vertical coverage (up to about 65 km altitude in model levels versus up to 50 km altitude in pressure levels) even though this implies an inhomogeneous upper limit of the profiles (which vary with geolocation in model levels). Each MIPAS scan is geolocated assuming vertical alignment of the tangent points of the sweeps at the center of the scan. In order to obtain the best coincidence between the geolocation of
MIPAS scans and ECMWF data, the assignment of ECMWF profiles to each MIPAS scan is performed selecting the ECMWF



6-hourly reanalysis file closest in time to the median scan of the orbit and adopting the model profile at the grid point nearest to the geolocation of the scan. Only one ECMWF file is adopted per orbit, even though the orbit may cross two ECMWF 6-hourly fields: this is to avoid introduction of spurious gradients passing from one ECMWF file to the following one within the same orbit. The adopted approach avoids interpolation and implies an uncertainty on the coincidence of $\pm 3h$ and $\pm 0.35$ degrees

latitude and longitude ($< 38\ \mathrm{km}$), which is consistent with typical validation activities.

Since the errors within the time and spatial window of coincidence are homogeneously distributed, this approach does not introduce any bias in the initial guess profiles.

## 5    Retrieval of minor species

The number of molecules retrieved by V8 L2 processor was increased with respect to previous processings, adding: $C_2H_2$,

$C_2H_6$, $CH_3Cl$, OCS, HDO and $COCl_2$ to the species provided in V7 data set, namely $H_2O$, $O_3$, $HNO_3$, $CH_4$, $N_2O$, $NO_2$, $CFC-11$, $CFC-12$, $N_2O_5$, $ClONO_2$, $COF_2$, $CF_4$, $HCFC-22$, HCN, $CCl_4$. The retrieval of the new trace species is made difficult by the weakness of their signal in MIPAS spectra and by the interference of other molecules. For this reason their analysis was only considered in the latest phase with an algorithm containing additional functionalities.

### 5.1    Optimal Estimation approach

For the full exploitation of MIPAS measurements, the retrieval of minor species was considered. For the trace species for which information is limited to a restricted altitude range or particular latitude regions, the regularizing Levenberg-Marquardt approach and a posteriori regularization method may provide an insufficient constraint. For this reason the Optimal Estimation technique (Rodgers, 2000) was introduced for the analysis of these species to avoid unphysical results.

The retrieval equation at the $i^{th}$ iteration was, therefore, modified with the addition of the optimal estimation constraint:

$$\boldsymbol{x}_{i+1} = \boldsymbol{x}_i + \left(\mathbf{K}_i^T \mathbf{S}_y^{-1} \mathbf{K}_i + \mathbf{S}_a^{-1} + \lambda_i \mathbf{D}_i\right)^{-1} \left[\mathbf{K}_i^T \mathbf{S}_y^{-1} (\boldsymbol{y} - \boldsymbol{F}(\boldsymbol{x}_i)) + \mathbf{S}_a^{-1}(\boldsymbol{x}_a - \boldsymbol{x}_i)\right] \tag{1}$$

where $\boldsymbol{x}_i$ is the retrieval vector at the $i^{th}$ iteration, $\boldsymbol{x}_a$ the a priori retrieval vector, $\boldsymbol{F}(\boldsymbol{x})$ is the forward model, $\mathbf{K}_i = \frac{\partial \boldsymbol{F}(\boldsymbol{x}_i)}{\partial \boldsymbol{x}_i}$ is the Jacobian, namely the derivative of the observations with respect to the quantities to be fitted, $\mathbf{S}_y$ is the CM of the observations, $\mathbf{S}_a$ is the CM of the a priori profile, $\lambda_i$ is the Levenberg-Marquardt parameter and $\mathbf{D}_i$ is the diagonal matrix with values equal to the diagonal of $(\mathbf{K}_i^T \mathbf{S}_y^{-1} \mathbf{K}_i + \mathbf{S}_a^{-1})$. When the a-priori is not considered $\mathbf{S}_a^{-1}$ is equal to zero and Eq. (1)

reduces itself to the formula used for the Gauss-Newton iteration modified with the Levenberg-Marquardt method.

During the iterations the Levenberg-Marquardt parameter is different from 0, but after convergence is reached, only if optimal estimation has been used, $\lambda_i$ is set to 0, an additional iteration is done and no a posteriori Tikhonov regularization is made. As a consequence, the CM and AKM do not have to take into account the Levenberg-Marquardt parameter and are given by:

$$\mathbf{S} = \left(\mathbf{K}_{i\_end}^T \mathbf{S}_y^{-1} \mathbf{K}_{i\_end} + \mathbf{S}_a^{-1}\right)^{-1} \tag{2}$$





$$\mathbf{A} = \left( \mathbf{K}_{\text{i\_end}}^{T} \mathbf{S}_{y}^{-1} \mathbf{K}_{\text{i\_end}} + \mathbf{S}_{a}^{-1} \right)^{-1} \mathbf{K}_{\text{i\_end}}^{T} \mathbf{S}_{y}^{-1} \mathbf{K}_{\text{i\_end}} \tag{3}$$

where i_end indicates the final iteration, with Marquardt parameter set to zero.

The a priori profile can be either the initial guess profile or a fixed profile, read from an external file, defined as the average of the climatological profiles of the considered trace species for all the seasons and all the latitude bands. The second option guarantees that the observed variability in the retrieved products comes from the measurements and not from the variability of the a priori profile. Only for the HDO retrieval, another possibility is foreseen, namely the a priori profile can be the $H_2O$ profile, retrieved in the same scan, scaled for the natural isotopic ratio. Indeed, the use of a fixed climatological profile for HDO would be critical due to the fact that the altitude of the hygropause can change significantly according to latitude and season. The a priori of continuum and offset are given by their initial guess, equal to 1 and 0 respectively.

The a priori CM is computed as follows: the diagonal elements of the sub-matrix related to VMR is calculated as a given percentage of the a priori profile plus an absolute error, useful when the a priori profile goes to zero and hence the constraint would become too strong. The off-diagonal elements of the sub-matrix related to VMR are calculated considering correlations decreasing exponentially with altitude differences, the strength of the correlations being tuned through a given correlation length (typically between 4 and 10 km). The sub-matrices related to the continuum and offset are diagonal calculated with an absolute error, equal respectively to one tenth of the maximum value of the continuum and $31.6\ nW/(cm^2 * sr * cm^{-1})$.

The use of the optimal estimation approach is selectable by input and it has been used only for the retrieval of the following trace species: HCN, CFC$-14$, COCl$_2$, CH$_3$Cl, C$_2$H$_2$, C$_2$H$_6$, OCS, HDO.

The profile retrieved with Optimal Estimation is the weighted mean of the information coming from the measurements and the information coming from the a priori, where the weights are given by the inverse of the CMs of the measurements and of the a priori profile, respectively. The result can hence be biased towards the a priori if the CM of the a priori is not sufficiently large. In order to limit the bias, a rather large CM is used for the a priori profile, its error being a percentage not smaller than 80% of its value. Nevertheless, for some constituents only few degrees of freedom are obtained from each retrieval and the a priori contributes significantly to the retrieved profile, so that useful results can only be obtained by averaging several observations. However, in the mean of many retrieved profiles, each containing the same a priori information, the a priori contributes with a reduced error and does no longer satisfy the implicit assumption of being, within the error, a realistic estimate of the real profile.

In the case of a large number of observations the averaging of the profiles is an important and frequent operation and the problem posed by the a priori is the reason why in the early analyses of MIPAS only the major atmospheric constituents, which did not require a priori information for the retrieval, were considered. Meanwhile a new method, in which the a priori components can be reduced, has been developed for the calculation of average profiles (Ceccherini et al., 2014). This new method removes the downsides of the use of a priori information, provided that information about the used a priori is made available to the users together with the products. In perspective, the possibility of changing the strength of the constraint in subsequent analyses is establishing new requirements for the provision of retrieval products.



## 5.2 Multi-target approach

In addition to the optimal estimation approach, in the new version of ORM the Multi Target Retrieval (MTR) functionality
(Dinelli et al., 2004), that is the simultaneous retrieval of two or more species, has been implemented. The MTR functionality
was meant to cope with strong contamination of the analysed spectral region by the emission of molecules that are insufficiently
characterised. However, the sequential retrieval, the careful selection of the spectral region to be analysed and the use of spectral
masks have enabled to reduce the contribution of the interfering species to the point that the MTR functionality was only used
for dedicated analyses and not for the full mission reprocessing of the MIPAS V8 L2 database.

## 6 Reduction of the outliers


An outlier is a data point that differs significantly from other observations. An outlier may be due either to the real variability
of the measurement or it may indicate an error in the measurements (corrupted spectra) or in the retrieval procedure (either
little information in the measurements which leads to large errors in the retrieved products and to instability of the retrieval
due to the ill conditioning, or unaccounted effects in the forward model). The challenge is to maintain the first type of outliers
and to filter out the others. Considering that the products of a retrieval can be used in subsequent retrievals, it is crucial to
assess the quality of the data and perform the filtering avoiding further error contamination. A large effort was dedicated to the
reduction of the outliers, both preventing the retrieval to be unstable, through the use of an updated approach for filtering out
cloud contaminated spectra, and after the retrieval, with a more comprehensive method to judge the quality of the products.

### 6.1 Improved cloud-filtering

Clouds impact the whole MIPAS spectral range with both smooth (cloud continuum) and sharp (line distortions due to scat-
tering) features. Both effects can be observed at the MIPAS spectral resolution (Spang et al., 2004; Greenhough et al., 2005;
Höpfner et al., 2006). Spectral signatures of polar stratospheric clouds can also be observed in MIPAS spectra (Höpfner et al.,
2018). This means that, while on one hand cloud information can be extracted from MIPAS measurements, on the other hand
cloud radiative effects can mask the spectral features of the target gases, strongly affecting their retrieval. As a consequence,
limb measurements affected by optically thick clouds must be identified and flagged, so that the retrieval of the trace gases
concentration can be performed using only the measurements with tangent height above the cloud. A cloud detection scheme
(Spang et al., 2004; Raspollini et al., 2006) is used to filter out spectra spoiled by clouds. A cloud index (CI) is computed
as the ratio between the mean radiance in the 788–796 $cm^{-1}$ interval, dominated by $CO_2$ and weak ozone emissions, and
the (832–834 $cm^{-1}$) interval, relatively insensitive to temperature, and characterised by aerosols and cloud emissions, as well
as some weak ozone and $CFC-11$ emission lines. Typical CI values for the upper troposphere are: CI = 1.8 corresponding
to thick cloud, CI = 6.0 meaning no cloud and 1.8 < CI < 4.0 corresponding to clouds of intermediate optical thickness. CI
values between 4 and 6 are produced by weaker cirrus clouds such as sub-visible cirrus clouds, or by clouds partially covering
the instrument field of view. CIs are computed for all the spectra of each scan with tangent height lower than 44 km. CIs are



then examined starting from the highest altitudes, using the tangent altitude of the first spectrum with CI smaller than a given

threshold to identify the cloud top height, and removing from the analysis all the limb views below it.

The CI threshold equal to 1.8 used in MIPAS/ESA processing up to V7 is not very effective at filtering polar stratospheric clouds (PSCs). Indeed, unfiltered PSCs were found to be responsible for the presence of outliers in the retrieved $H_2O$, $NO_2$ and $N_2O_5$ for winter time in the southern polar region. Sometimes the used CI threshold poorly filters high-altitude tropical clouds, thus producing unrealistically large VMRs (e.g. $CH_4$) in these regions. As expected, on the basis of the increased opacity of the

atmosphere due to the presence of clouds, the profiles containing outliers are also characterised by large retrieval errors. Figure 17 shows, as an example, the time series of the monthly mean profiles of $H_2O$ VMR (upper panel) and of its retrieval errors (bottom panel) for a CI threshold of 1.8. The maps refer to the latitude belt from 60 to 90°S. Outliers are clearly visible in the polar winter of each year between 10 and 60 hPa, where polar stratospheric clouds may occur. Correspondingly we observe very large retrieval errors, confirming the reduced sensitivity of the measurements due to the opacity of the clouds.

Figure 18 shows examples of the CI profiles computed for all limb scans of orbit 33153 from 3 July 2008. On the same figure the CI threshold of 1.8 (used in the MIPAS/ESA L2 analyses up to V7) and a more conservative value equal to 4.0 (respectively also sweeps affected by relatively transparent clouds and used in other independent analyses like in Dinelli et al. (2010) and von Clarmann et al. (2009)) are reported. We see that most scans are characterised by CI values larger than 6 above $\approx 26$ km. In general, the CI decreases while moving to the lowest altitudes, however, more abrupt reductions are visible in the presence

of a cloud. In Figure 18, the two thicker lines with dots (dark blue and red) represent two specific limb scans for which the use of a constant CI threshold for cloud flagging does not produce optimal results. The dark blue line relates to scan 71 measured at a latitude of 61°S (polar winter), its CI profiles indicates the presence of a cloud (a polar stratospheric cloud) with top height around 24 km, thus all sweeps below this altitude should be excluded from the analysis to safely retrieve good quality VMR profiles. The red line relates to scan 22 measured at 63°N latitude (polar summer), in this case the CI profile decreases

smoothly when moving to lower altitudes, therefore this scan should be classified as clear and all the limb views could be used in the inversion. The water vapour profiles retrieved from scans 22 and 71 are shown on the left and right panels of Fig. 19, respectively. More specifically, Fig. 19 shows the water profiles retrieved from the clear limb views selected using fixed CI thresholds of 1.8 (red profiles) and 4 (blue profiles). From Fig. 19, we clearly see that the threshold of 1.8 is not adequate for scan 71, as it leaves unfiltered the sweep with tangent height around 24 km, responsible for the unrealistically large $H_2O$ VMR

at the same altitude (see the red profile on the right panel of Fig. 19). For this scan the CI threshold equal to 4 performs much better as it correctly filters all the sweeps affected by clouds, thus permitting a reasonable retrieval (see the blue profile on the right panel of Fig. 19). On the other hand, a CI threshold of 4 is too conservative for scan 22 as it erroneously flags as cloudy the two lowest sweeps of the limb scan (see the blue and red profiles on the left panel of Fig. 19). The ORM processor version 8 overcomes these difficulties by using altitude- and latitude- dependent CI thresholds obtained by multiplying by 0.8 the

thresholds defined in Sembhi et al. (2012) and Griessbach et al. (2018). The factor 0.8 stems from the fact that the thresholds reported in the above mentioned papers were developed to derive information on clouds from the measurements, while our goal is only to filter the clouds that significantly impact the retrievals. Examples of CI threshold profiles for polar and tropical latitudes are shown in Fig. 18 with the solid thick blue and magenta lines, respectively. As can be easily seen from Fig. 18, the

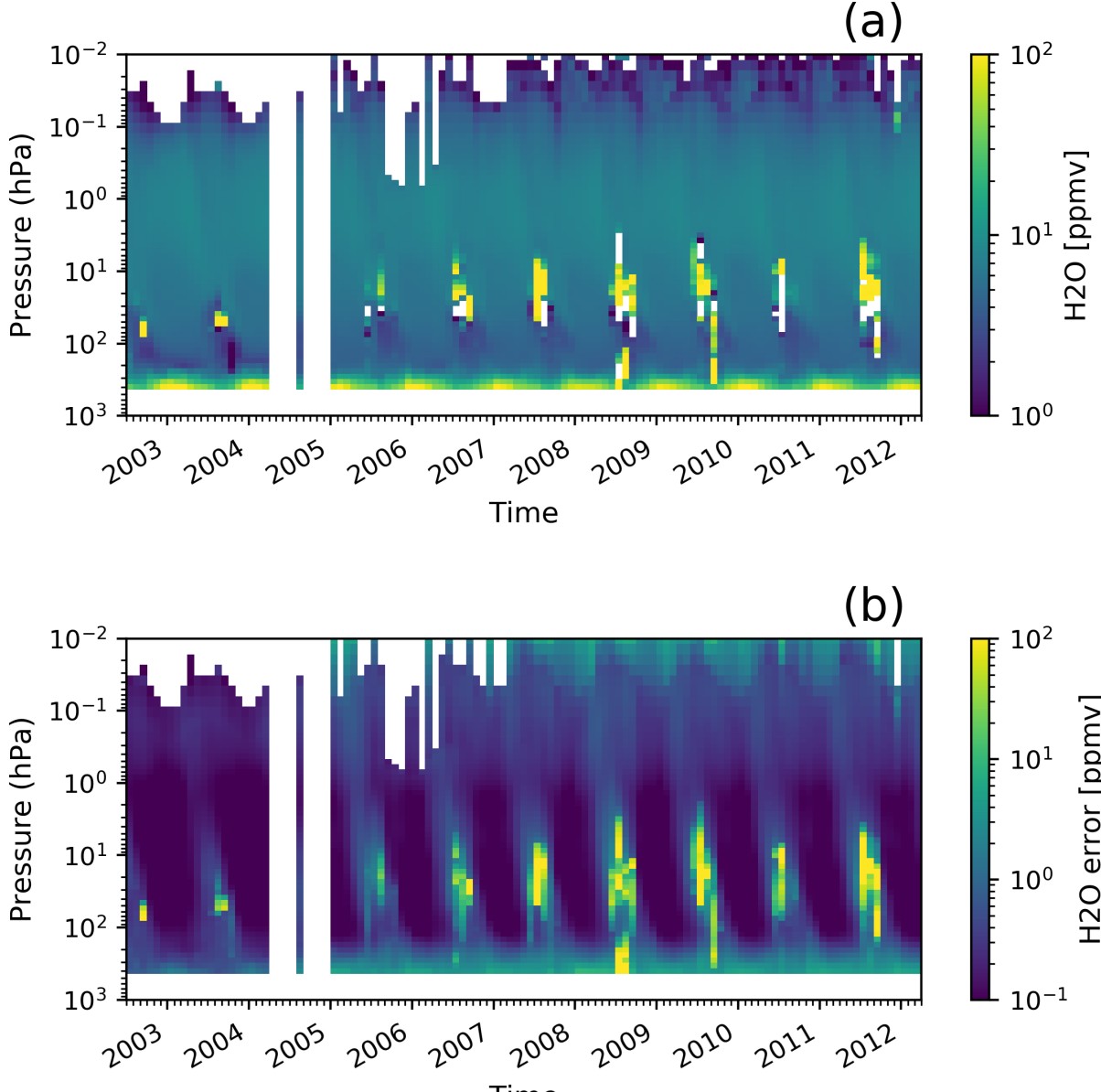

**Figure 17.** (a) Time series of $H_2O$ monthly mean profiles in the $60 - 90°$ S latitude belt and (b) corresponding mean single scan random error profiles when using a CI threshold of 1.8.




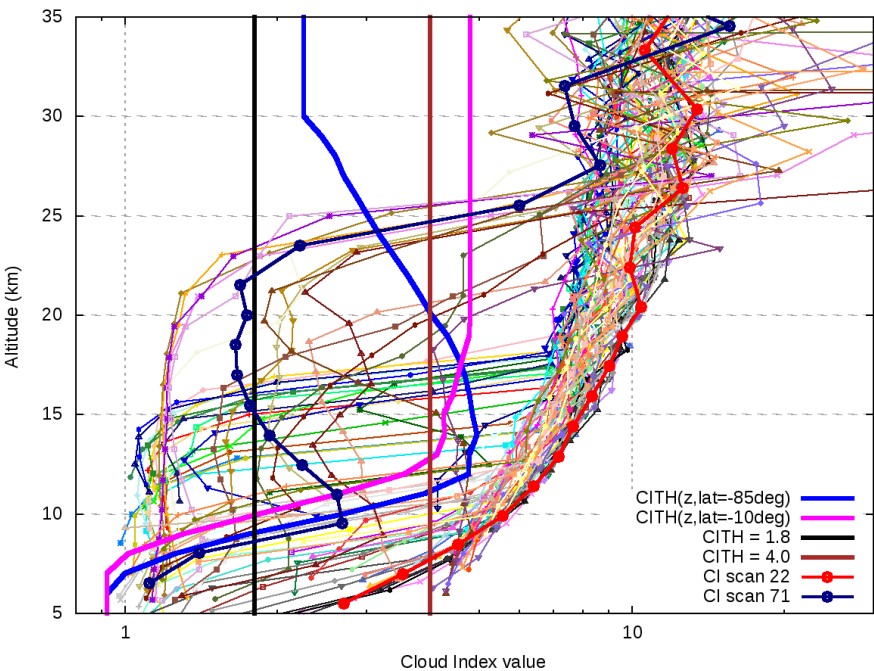

**Figure 18.** CI profiles computed for all limb scans of orbit 33153 from 3 July 2008. The thicker lines with dots, dark blue and red respectively, refer to two selected scans at latitudes 61°S (polar winter), scan number 71, and 63°N (polar summer), scan number 22. The water vapour profiles retrieved from these two scans are shown in Fig. 19.

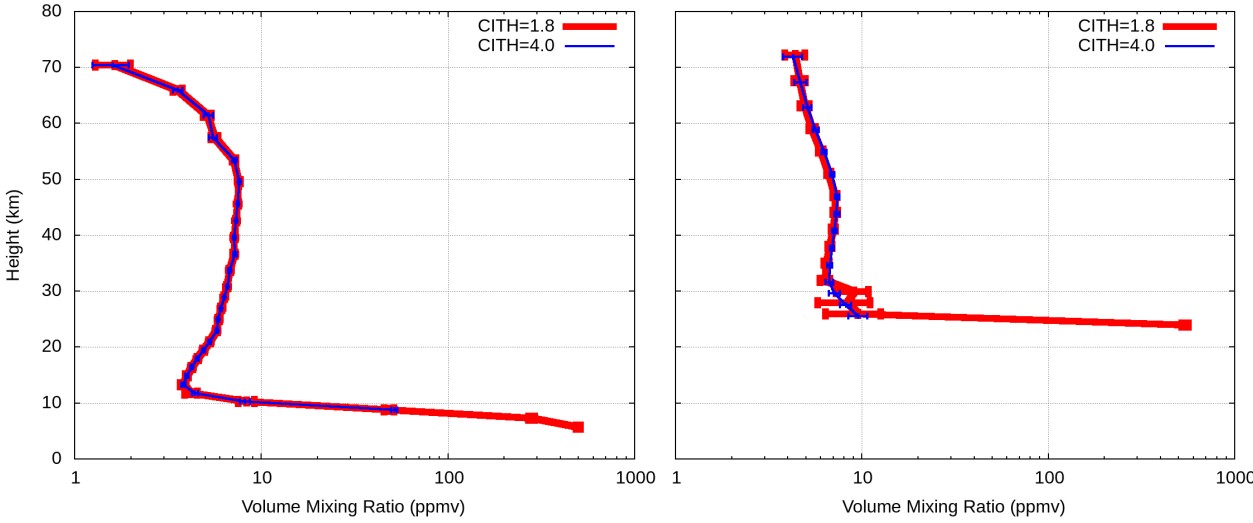

**Figure 19.** Water vapour profiles retrieved from scan 22 (left panel, 63°N) and scan 71 (right panel, 61°S) of orbit 33153 from 3 July 2008. Red profiles are retrieved after filtering the measurements with a CI threshold of 1.8, the blue profiles with a CI threshold of 4.0.



polar CI threshold profile permits to properly flag the stratospheric cloud that affects scan 71, and to retain all the clear sweeps

of scan 22. As expected, the altitude- and latitude- dependent cloud filtering has the effect of significantly reducing the number of outliers in polar winter profiles. Figure 20 is a time series of $H_2O$ monthly mean profiles (upper plot) and errors (bottom plot) retrieved with the ORM V8, from measurements in the $60 - 90°S$ latitude belt. Comparing these maps with Fig. 17, we see how the outliers in both profiles and errors have been removed by the new cloud filtering.

With the new altitude- and latitude- dependent CI thresholds, the statistics of the number of retrieved profile grid points

changes with respect to Level 2 V7 products. Table 2 shows the percentage variation (V8 vs V7) in the number of retrieved temperature profile grid points for different height and latitude ranges. The statistics reported in the table is based on a sample of $\approx 4000$ orbits of nominal (NOM) and UTLS1 measurements of both FR and OR mission phases. The increased number of V8 retrieved data points above $30\,\mathrm{km}$ arises from the fact that the a posteriori filtering (see Sect. 6.2) actually removes a smaller number of profiles containing outliers. In the intermediate range, from 7 to $15\,\mathrm{km}$ the V8 cloud masking strategy seems

far more effective than V7 one, especially in polar and tropical regions. At the lowest altitudes, the new cloud filtering is less conservative than in the V7 data set, thus, the number of retrieved profile points is greater in the V8 data set.

Errera et al. (2016) found a very poor agreement between MIPAS V6 and V7 products and colocated products of MLS and ACE-FTS, especially in the Tropical lower stratosphere. Since the discrepancies were mainly due to MIPAS outliers, we expect an improved agreement when the V8 data set are considered. As an example, Fig. 21 compares the V7 and V8 time series of

the Tropical $CH_4$ VMR monthly means at the pressure level of $82\,\mathrm{hPa}$. As we can see, the V7 data set contains a lot of outliers, especially in the FR part of the mission, that are no longer present in the V8 data set. In some cases the outliers in the V7 data also hide the seasonal variation of the target parameter considered. This seasonality is now correctly represented in V8 data.

## 6.2 Quality flagging

In the previous section we have discussed how, by better filtering out spectra affected by clouds, it is possible to reduce

the number of outliers in the retrieved profiles. However, outliers can also be the result of problems due to the presence of corrupted spectra or to problems in the iterative retrieval procedure. These can be identified through a quality control of the products performed after the retrieval. In order to avoid any bias in the averages of the retrieved profiles, no constraints are imposed on the values obtained at the end of the retrieval procedure. The quality of the retrieved profiles is judged 'good' when three requirements are met: the retrieved profile adequately reproduces the measurements (judged using the reduced

chi-square), there are no outliers in the retrieval error and the iterative retrieval procedure successfully converges.

The first condition is met when the chi-square value at the final step of the iterative procedure is smaller than a pre-defined mode and species dependent threshold. The chi-square provides a measure of how well the results of the retrieval are able to reproduce the measurements within the measurement errors. In absence of any systematic error or retrieval instability, the spectral residuals should be within the measurement noise level and the expected value of the reduced chi-square equals 1.

Although a high chi-square does not necessarily imply that the retrieved quantities are not realistic, exceptionally high values of the chi-square were found to be associated with bad quality profiles.

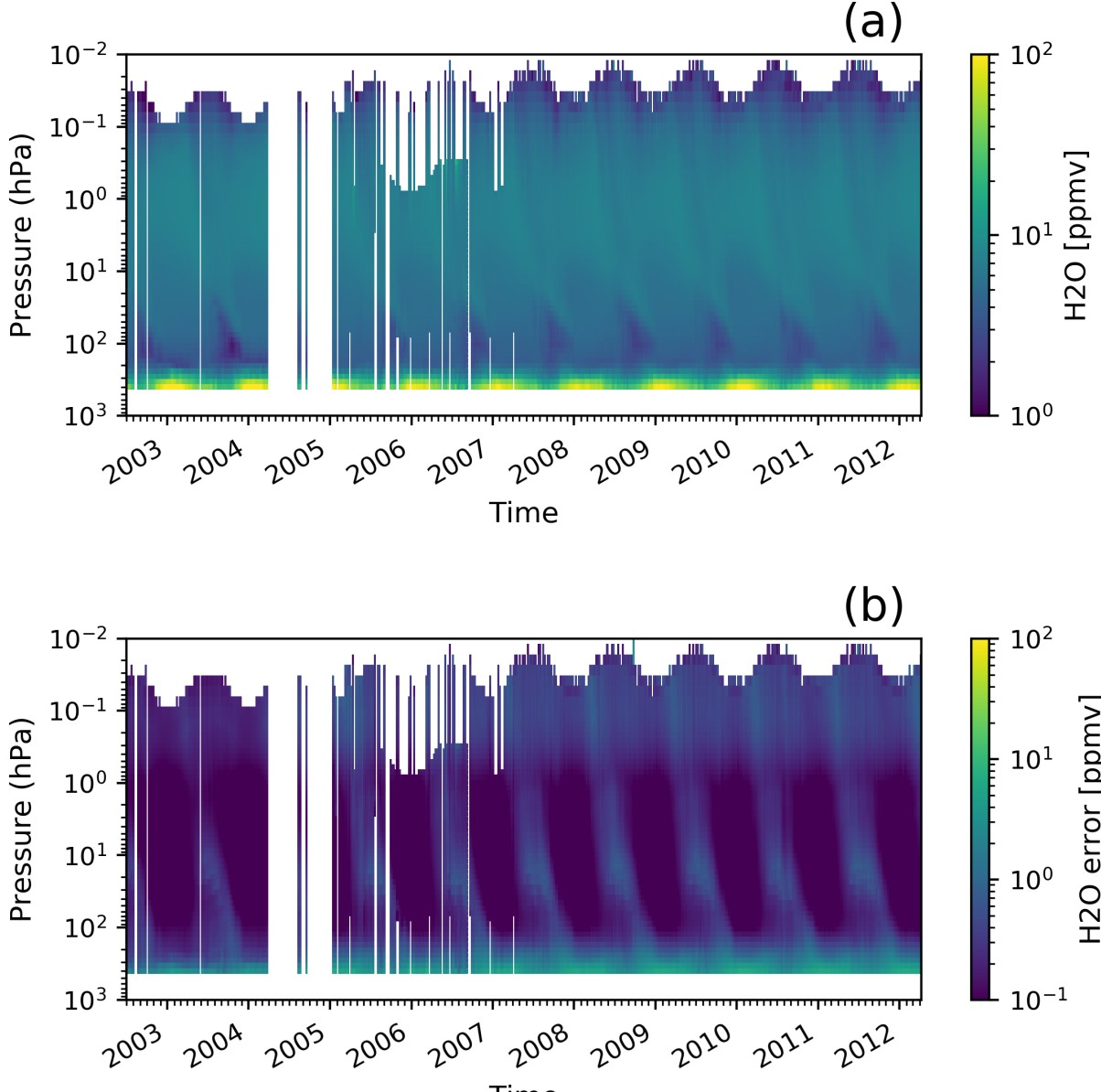

**Figure 20.** (a)Time series of monthly mean V8 $H_2O$ profiles in the $60 - 90°$S latitude belt and (b) corresponding mean single scan retrieval error profiles when latitude and altitude dependent CI profiles are used.





| case FR nominal | | | | | | |
|---|---|---|---|---|---|---|
| | **90S-60S** | **60S-30S** | **30S-0** | **0-30N** | **30N-60N** | **60N-90N** |
| **30-99 Km** | 2.0% | 0.6% | 1.8% | 1.6% | 0.1% | 0.0% |
| **15-30 Km** | -4.0% | -0.2% | -1.9% | 2.3% | -0.7% | -2.0% |
| **7-15 Km** | -10.2% | -4.3% | -12.8% | -15.6% | -4.4% | -5.0% |
| **0-7 Km** | 64.1% | 84.5% | 79.8% | 68.4% | 111% | 83.4% |
| case OR nominal | | | | | | |
| | **90S-60S** | **60S-30S** | **30S-0** | **0-30N** | **30N-60N** | **60N-90N** |
| **30-99 Km** | 1.8% | 0.7% | 1.4% | 0.6% | 0.0% | 0.6% |
| **15-30 Km** | -5.1% | 0.6% | -2.6% | -3.6% | -0.3% | -1.6% |
| **7-15 Km** | -15.3% | -6.1% | -19.0% | -23.0% | -11.5% | -16.4% |
| **0-7 Km** | 56.1% | 67.0% | 50.0% | -13.3% | 88.6% | 66.7% |
| case OR UTLS1 | | | | | | |
| | **90S-60S** | **60S-30S** | **30S-0** | **0-30N** | **30N-60N** | **60N-90N** |
| **30-99 Km** | 2.3% | 0.6% | 3.0% | 4.2% | 0.5% | -0.4% |
| **15-30 Km** | -4.4% | 0.1% | -1.3% | -1.0% | 0.5% | -0.9% |
| **7-15 Km** | -12.3% | -7.4% | -17.0% | -21.4% | -6.1% | -7.1% |
| **0-7 Km** | 49.1% | 880% | 0.0% | 0.0% | 203% | 77.8% |

**Table 2.** Percentage differences in the number of temperature profile grid points retrieved with the V7 – V8 CI thresholds. The statistics is given, separately, for four height ranges and six latitude bands. It includes a sample of ≈4000 orbits acquired in Nominal (both FR and OR) and UTLS-1 modes.

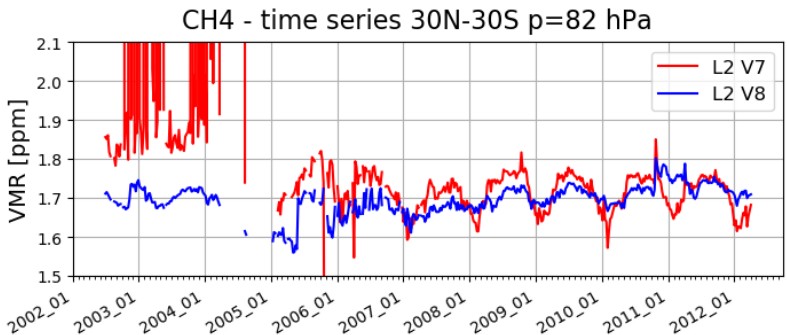

**Figure 21.** Time series of $CH_4$ monthly mean profiles in the Tropics, at the pressure level of 82 hPa.



The second condition on retrieval errors is met when the maximum value of the retrieval error profile is smaller than a pre-defined mode and species dependent threshold: indeed, it has been found that there is correlation between outliers and very large retrieval errors.

The third condition is met when convergence of the Gauss Newton iterative procedure is attained. This happens if at least one of the following criteria is verified:

- The relative variation of the chi-square between two consecutive Gauss iterations is smaller than a pre-defined threshold (set to 0.01). This is verified in the large majority of the scans.

- The weighted $L^2$ norm of the difference between the vector of the retrieved profiles at two consecutive iterations, nor-
malized with the number of retrieved points of the profiles, is smaller than a given threshold. The weight is given by the inverse of the CM of the retrieved parameters. It guarantees that convergence is reached when the retrieval parameters do not change even if there is some variation in the chi-square. The threshold is set to 0.1 for the pT retrieval, and 0.08 for the VMR retrievals. Only a small number of cases fulfils this criterion.

On the other hand, the retrieval iterations are stopped and convergence is not attained if one of the following two conditions
is verified: the maximal number of Gauss iterations (set to 15) is reached; the maximum number of consecutive Marquardt micro-iterations (set to 5) is reached. We are in the latter condition when, even with a large Levenberg-Marquardt parameter, no reduction of the chi-square is possible. The successful convergence also requires that the final value of the Marquardt parameter is smaller than a given threshold. Indeed, a too large value of the Marquardt parameter, even if properly taken into account for the computation of the CM and AKM, implies a very small retrieval step and hence it may trigger false convergence.

When at least one of the previous requirements is not verified, the retrieved profile is flagged as 'not-good' in the output file (post_quality_flag=1) and it is not used as either profile of an interfering species or initial guess in subsequent retrievals. Otherwise, if all previous conditions are verified, the post_quality_flag is set to 0 so that the retrieved profile is considered 'good' and it can be used for subsequent retrievals. If the retrieved temperature is flagged as not-good, no VMR retrieval is performed, since a proper temperature profile is fundamental for the retrieval of the trace species. The retrieved profiles of
some minor species (namely $COF_2$, $CCl_4$, $CF_4$, $HCFC-22$, $C_2H_2$, $CH_3Cl$, $COCl_2$, $C_2H_6$, OCS, HDO) are not used in the subsequent retrievals of the chain even if the quality of the products is 'good'. This choice was based on a prudence principle, considering that the improvements coming from taking the interference of these trace species in other retrievals are very small, while the presence of possible outliers may impact negatively the retrievals.

The thresholds for chi-square and maximum error for each species and measurement mode were defined a posteriori using
the results of the analyses performed on the so-called Diagnostic Data set. This consists of about 4000 orbits, e.g. about one tenth of the orbits of the full mission, selected to cover the whole mission period in coincidence to most of the correlative measurements used for validation.

Figure 22 shows an example of the chi-square distribution of a representative retrieval. The percentile level Lx of the distribution, with 0<x<100, is determined as the value of the sample such that x% of the total number of samples is less
than Lx. The value of chi-square corresponding to the maximum of the histograms has been labelled as 'Hmax'. The criteria





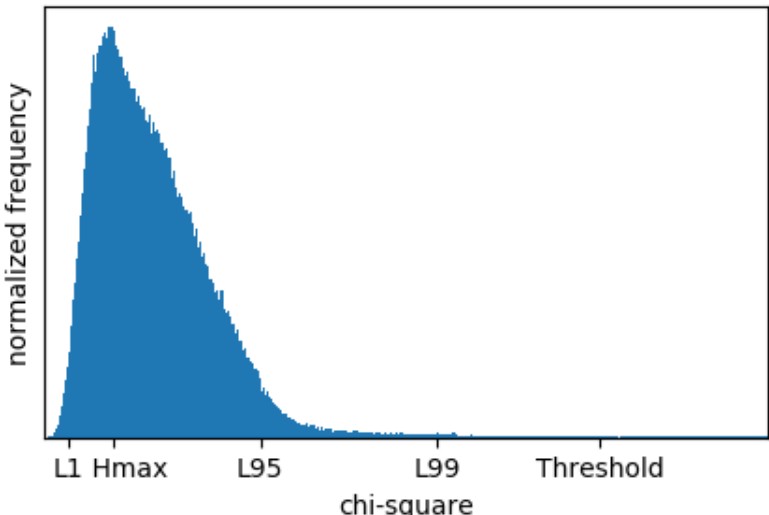

**Figure 22.** Example of chi-square distribution and considered statistical levels

used to define the thresholds have been chosen empirically trying to satisfy the two conflicting requirements of filtering out the largest number of outliers and of not throwing away 'good' profiles. Considering the different shape of the distributions of chi-square and errors, the values of the thresholds have been then determined using different values of percentile as follows:

CHI-square_threshold= L99+(L99-Hmax)*0.5,

MaxError_threshold= L95+(L95-Hmax).

The chi-square distribution is reported in orange in Figs 1 and 2 for some species considering all the profiles retrieved from the nominal measurement mode, respectively for the FR and the OR phases of the mission. The full-scale value of the x axis is given by the selected threshold. It is evident that only the very far right tail of the distribution is filtered out. The distributions of V7 products are overlapped in blue for a comparison. The maximum error distribution (not shown) has a similar shape as

the chi-square distribution.

The used thresholds for chi-square, maximum error and final Marquardt parameter are given in the output NetCDF files (see Dinelli et al. (2021)).

## 7 Improvements in temporal stability of the measurements

Ten years of measurements may be a short period to derive trends, but surely they can be used to study the time evolution

of many species. Furthermore, these measurements can be used in combination with measured data from other sources. The potentiality to study the time evolution of the species that can be retrieved from MIPAS measurements with high accuracy requires a very stable instrument and a careful elimination of any instrumental drift. V8 of the L1b processor (Kleinert et al.,



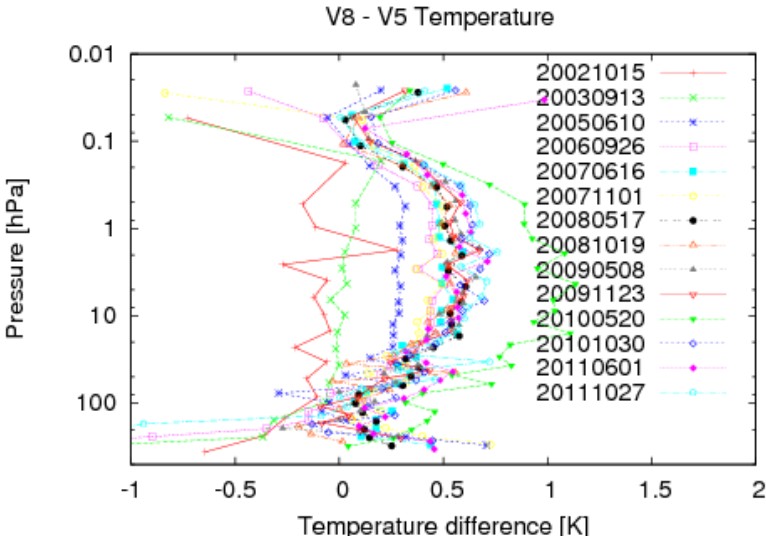

**Figure 23.** Differences in temperature daily means between V8 and V5 L2 data for 14 selected days (October 2002 to October 2011). These differences are small at the beginning of the mission but they gradually increase with time.

2018) includes several improvements as compared to earlier versions which have an impact on the temporal variation of the retrieved products. The impact of these improvements on the L2 products has been analysed.

## 7.1 Improvements in the non-linearity correction

The MIPAS photoconductive detector channels A1/A2, B1/B2, measuring the longest wavelengths, show a non-linear response to the incident photon flux, which has to be taken into account by the L1 processor prior to radiometric calibration. This can be done only if the Non-Linearity (NL) curve, representing the detector response (voltage) as a function of the incident photon flux, is available. NL correction parameters measured on ground before launch were applied to all the measurements up to L1b V5 products. During the mission, in order to reveal possible changes in the NL and to improve its characterisation, dedicated measurements were acquired. The analysis of the in-flight characterisation measurements revealed that, because of the aging, the photoconductive detectors response slowly decreases, and the detectors become more linear over time. Moreover, the characterisation showed that the NL detector curve depends on the instrument temperature and on the degree of ice contamination (Birk and Wagner, 2010). As a consequence, new parameters for the NL correction were determined from in-flight characterisation measurements. The various upgrades implemented in the NL correction parameters have an impact on the L1b products which in turn affect the L2 products and in particular the trends that may derived from these. Figure 23 shows, for selected days all over the mission, the difference between daily averages of temperature profiles retrieved from either V8 or V5 L1b products. The profiles retrieved at the beginning of the mission are very similar for the two versions but the average differences gradually grow during the mission, since the new NL correction increases with time.





A comparison of the MIPAS instrument drift for different versions of the MIPAS L2 products, computed relative to ra-
diosonde and lidar, is presented by Hubert et al. (2020). According to their Fig. 33 a significant reduction in temperature
drift between 500 hPa and 0.1 hPa is found for version V8 data in the OR period. A small residual positive drift of at most
0.4 K/decade below 10 hPa still remains. The residual drift can be explained by the uncertainty of the NL correction, mainly
due to the fact that the measurements for its characterisation are sparse (only 35 orbits are available throughout the whole

mission) and, even more important, the continuum level of the signal, which is needed to place the measurements correctly
on the NL detector curve, is not provided and has to be estimated from the measured AC-coupled interferogram. The residual
uncertainty on the time-dependency of the radiometric error has been estimated to be less than 0.5%/decade (see Fig. 11 of
Kleinert et al., 2018). It has to be noted that this uncertainty may have a different impact on the various trace species due to the
different spectral intervals selected for the analysis; indeed species with spectral features in the bands at the lowest frequencies

are more affected by the NL problem.

## 7.2   Improved gain calibration

Errera et al. (2016) reported discontinuities up to 7% in the time series of the daily means of both V6 and V7 $CH_4$ and
$N_2O$ MIPAS data, occurring shortly after decontamination periods. Decontamination consists in a warming of the instrument
to remove ice from the mirrors. The discontinuities in the time series of $CH_4$ and $N_2O$ were found to be correlated with

unexplained abrupt changes of up to 2% in the gain of band B (Kleinert et al., 2018) where the spectral lines of $CH_4$ and
$N_2O$ used for the retrieval are located. The radiometric gain is the factor the observed scene is multiplied by to obtain the
calibrated radiance. Before this operation, the observed scene is corrected subtracting the deep space measurement closest in
time for taking self-emission of the instrument into account. The gain is determined from a series of blackbody and deep space
measurements on a daily basis. However, until L1b V7 products the gain function used for radiometric calibration was updated

only once per week. Indeed, the gain variation is usually sufficiently slow, so that the error introduced by the temporal drift
of the gain function was expected to be small, and weekly averages were chosen to reduce the noise error on the calibration.
However, if the gain changes in the time frame of 1 or 2 days and if the radiometric calibration is not performed at the
corresponding days, a discontinuity in the radiance occurs that translates in a discontinuity in the time series of the retrieved
products. It has been proven with dedicated tests that, since the relation between measured radiance and retrieved profile is not

linear, a change of 2% in the gain can induce a change of up to 7% in the retrieved profile of $CH_4$, as found in the time series.
V8 L1b products were generated with gain measurements updated every day and this allows to better capture the gain variation
and hence to reduce the discontinuities in the VMR time series.

    Figure 24 shows in red the $CH_4$ global average time series for September 2003 derived from L2 V7 MIPAS data at a pressure
of 49.8 hPa. Also the used gain function is reported, in green, with the triangles indicating the days at which the gain has been

updated (approximately once per week). A large discontinuity occurs from the $9^{th}$ to the $13^{th}$ of September 2003 in the $CH_4$
VMR. Figure 25 reports in blue the same $CH_4$ global average time series derived from L2 V8 MIPAS data and in green the
corresponding gain values, which are daily updated. We see a strong variation in the gain in the period $9^{th}$ to $13^{th}$ of September
which had been completely masked by a weekly sampling of the gain. The handling of the gain variation allows to significantly

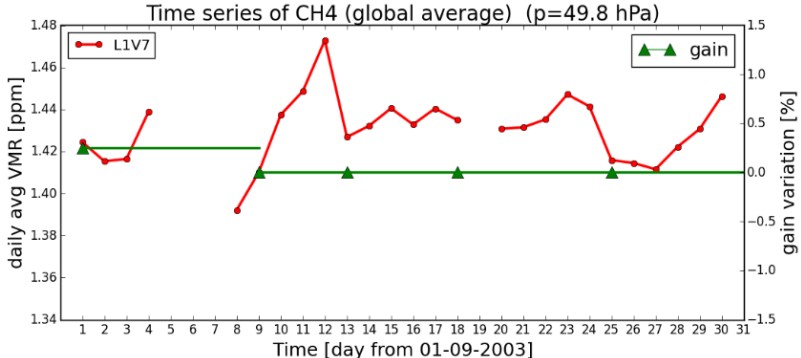

**Figure 24.** Time series of $CH_4$ global average at pressure 49.8 hPa relative to beginning of September 2003 derived from L2 V7 MIPAS data (red line) and the used gain function (green line) with the triangles indicating the days at which the gain function has been updated.

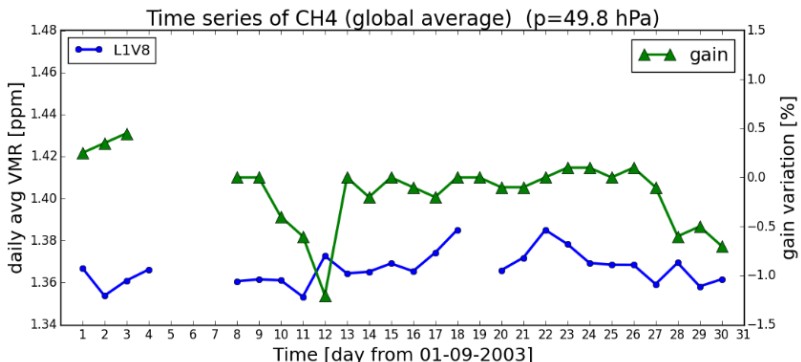

**Figure 25.** Time series of $CH_4$ global average at pressure 49.8 hPa relative to begin of September 2003 derived from L2 V8 MIPAS data (blue line) and the used gain function (green line) with the triangles indicating the days at which the gain function has been updated.

reduce, in L2 V8 data, the discontinuities which were present all over the mission in the time series of V6 and V7 $CH_4$ and
$N_2O$ MIPAS data.

## 8 New format for L2 products

The format adopted for the output files of MIPAS L2 V8 data set is netCDF-4. Two different kinds of files are released: a standard file and an extended one. The choice of providing two output files was driven by the need to make their use easier and to distinguish between information that is needed by most users and information that is needed for making diagnostics of
the products and for special analyses. Each standard file provides all the information that is generally needed for validating and exploiting MIPAS L2 products. Each file refers to a single orbit and to either a single trace species or temperature: it contains, for each scan of the orbit, its geolocation and the retrieved profile with related CM and AKM, as well as used altitude,



pressure and temperature profiles and their errors. The quality flags are also available to filter out 'non-good' profiles. The temperature file contains as additional information also the cloud index profile, from which it is possible to deduce the cloud

top height, as the altitude below the lowest tangent altitude included in the analysis. The main difference in the content of the standard and the extended files is that the extended file provides the full state vector and its related CM and AKM. The full state vector contains, in addition to the retrieved profiles (either VMR profile or pressure and temperature profiles), the atmospheric continuum profiles and instrumental offset. This information is useful when making data fusion (Ceccherini et al., 2015). Further information is provided in Dinelli et al. (2021).

**9   Conclusions**

The new version of the MIPAS ESA L2 processor ORM V8 allows the retrieval of the temperature profile and the VMR profile of 21 trace species: water vapour, ozone and many other longer-lived greenhouse gases ($CH_4$, $N_2O$, $CFC-11$, $CFC-12$, $CF_4$, $CH_3Cl$, $CCl_4$), nitrogen and halogen species of interest for ozone chemistry ($ClONO_2$, $COF_2$, $N_2O_5$, $COCl_2$, $NO_2$, $N_2O_5$, $HNO_3$), sulphur compounds (OCS), species contained in biomass burning and other pollution plumes (HCN, $C_2H_2$, $C_2H_6$),

isotopologues (HDO).

The L2 processor ORM V8 and its auxiliary data changed significantly with respect to the ESA L2 processor ML2PP V7 used to perform the previous full mission reanalysis. The modelling of the measurements was improved by taking into account the horizontal inhomogeneities along the line of sight. Horizontal gradients computed from ECMWF ERA-Interim reanalysis were assumed and this mainly led to a reduction of the ascending-descending differences, previously wrongly attributed to

daytime-nighttime differences, in profiles where a diurnal variation is not present.

The minor species (HCN, $CF_4$, $COCl_2$, $C_2H_2$, $C_2H_6$, $CH_3Cl$, HDO, OCS) were retrieved with the Optimal Estimation method allowing stable retrievals also when using large a priori errors. A better modelling of the measurements was obtained also through the use of improved spectroscopy. HITRAN_mipas_pf4.45 spectroscopic database was finalized for this full mission reprocessing. This contains spectroscopic data from different versions of HITRAN databases, data validated with MI-

PAS measurements themselves (and now contained in HITRAN 2016), new data. For the heavy molecules new cross-sections were used by ORM V8 for the following molecules: $CFC-12$, $CFC-14$, $HCFC-22$, $CCl_4$, $ClONO_2$, $HNO_4$, $CFC-11$, $CFC-113$ and $SF_6$. In general the use of an updated spectroscopy leads to a better representation of the observations, with smaller differences between observations and simulations, and to significant changes in some retrieved products (HCN, $HNO_3$, $CCl_4$, $CFC-11$, $CFC-12$, $HCFC-22$).

A large effort was dedicated to the representation of the atmosphere, needed for defining the initial guess of the retrieval state vector, the interfering species, the horizontal gradients and the a priori profiles for trace species retrieved with Optimal Estimation. Even if for the choice of the initial guess and interfering species the highest priority is given, when available, to the previously retrieved profiles in the same or previous scan, different databases were fine-tuned for the different scopes and used by the retrieval code according to a priority list. The IG2 V5.4 database represents an evolution of the database IG2 V4.0

described in Remedios et al. (2007), and includes climatologies of the daytime and nighttime atmosphere from 0 to 120 km



for numerous species. The ERA-Interim ECMWF data set was also used, after the identification of the closest coincidences for each MIPAS scan, for computing the horizontal gradients of temperature, water vapour and ozone. The improved modelling of the measurements was proven by a significant general reduction of the chi-square values for most of the species.

The cloud filtering, used to filter out spectra corrupted by clouds, is in this version latitude and altitude dependent and this
improves the sensitivity to the presence of clouds. The use of these new thresholds allowed to filter out spectra affected by polar stratospheric clouds, reducing significantly the number of outliers in water vapour and $N_2O_5$ retrieved profiles during the polar winter. In general the use of these altitude and latitude-dependent thresholds reduced the number of retrieved points in the UTLS, but increased the number of retrieved values below 15 km and above 30 km.

Other improvements in the V8 L2 products are a consequence of the reduced instrumental drift and of the use of measured
daily gain which allows a reduction of the discontinuities in the timeseries of some species. Output files are provided in NetCDF and contain the L2 products as well as all characterising quantities (CM and AK), and only a flag is needed to filter out low quality products. This flag takes into account various qualifiers which have been further refined and takes into account successful convergence of the retrieval iterations, the capability of the retrieval to reproduce the measurements and the presence of outliers in the retrieval error. The description of the implemented improvements is meant to better understand the products
of the ESA L2 V8 reprocessing (which are further characterised in terms of spatial coverage, retrieval errors, spatial resolution, 'useful' vertical range considering the whole L2 V8 data set in Dinelli et al. (2021)) and also to help developers of retrieval codes.

**Appendix A: The IAA data set**



**Table A1.** Summary of the changes introduced in the SLIMCAT output to generate the IAA data set.

| Gas | Data source |
|---|---|
| $N_2$ | MSIS[a] (up to 200 km). |
| $O_2$ | MSIS (up to 200 km). |
| $CO_2$ | IG database (V4) below 50 km. WACCM between 50 to 140 km. Extrapolated above up to vmr=2 ppmv at 200 km. |
| $H_2O$ | SLIMCAT below 65 km. Garcia and Solomon (1994) above up to 110 km. Extrapolated above to vmr=$10^{-4}$ ppmv at 200 km. |
| CO | SLIMCAT below 35 km. Garcia and Solomon (1994) above up to 110 km. Extrapolated above to vmr=1 ppmv at 200 km. |
| HCN | IG database (V4) up to 120 km. Extrapolated above to vmr=$4.5 \times 10^{-5}$ ppmv at 200 km. |
| OH | Garcia and Solomon (1994) up to 90 km. In photochemical equilibrium with H above 110 km (using IAA box model). |
| H | Garcia and Solomon (1994) up to 90 km. MSIS above up to 200 km. Low cut-off of $10^{-14}$ ppmv. Smoothing applied[†]. |
| $HO_2$ | SLIMCAT below 75 km. Extrapolated above with vmr=$10^{-6}$, $10^{-9}$, and $10^{-10}$ at 100, 140 and 200 km. Smoothing applied[†]. |
| $O_3$ | SLIMCAT below 60 km. Garcia and Solomon (1994) above (up to 110 km) with diurnal correction for MIPAS am and pm times (IAA box model) between 67-82 km. Extrapolated above with vmr=$10^{-6}$ and $10^{-8}$ at 140 and 200 km. Smoothing applied[†] above 75 km. |
| O | IAA box model below 80 km (consistent with $O_3$). MSIS above up to 200 km. Low cut-off of $10^{-15}$ ppmv. Smoothing applied[†]. |
| $O(^1D)$ | IAA box model. |
| $NO_2$[‡] | SLIMCAT below 20 km. Garcia and Solomon (1994) above up to 110 km, diurnal correction applied taking the $NO_x$ from Garcia and Solomon (1994) and $NO_2$/NO partitioning from the IAA box model. Above 110 km calculated with IAA box model. Smoothing applied[†] below 30 km and above 75 km. |
| NO[‡] | IAA box model (consistent with $NO_2$ as explained above) below 85 km. NOEM[b] model above up to 200 km. Low cut-off of $10^{-8}$ ppmv and smoothing applied[†] for computational stability in the calculation of vibrational temperatures. |
| N | IAA box model (consistent with NO) below 85 km, MSIS above (up to 200 km). |
| $CH_4$ | SLIMCAT up to 75 km. Extrapolated above with vmr=$10^{-9}$, $10^{-12}$, and $10^{-15}$ at 100, 140 and 200 km. |
| $N_2O$ | SLIMCAT up to 75 km. Extrapolated above with vmr=$10^{-9}$, $10^{-12}$, and $10^{-15}$ at 100, 140 and 200 km. |
| $HNO_3$ | SLIMCAT up to 75 km. Extrapolated above with vmr=$10^{-9}$, $10^{-12}$, and $10^{-15}$ at 100, 140 and 200 km. Smoothing applied[†] above 45 km. |
| ClO | SLIMCAT up to 75 km. Extrapolated above with vmr=$10^{-7}$, $10^{-14}$, and $10^{-16}$ at 100, 140 and 200 km. Smoothing applied[†] at all altitudes. |

[a] Picone et al. (2002). [b] Marsh et al. (2004). [†] A log-smoothing with a vertical length of 5 km has been applied to remove unphysical oscillations in the original profiles. [‡] The $NO_2$ and NO from the SLIMCAT model were modified below 80 km in order to account for the $NO_x$ descent in the polar regions. Outside the polar regions the SLIMCAT model and the Garcia and Solomon (1994) and IAA box models give very similar values.



*Data availability.* Spectroscopic database pf4.45 is available at the following web-page:

https://earth.esa.int/eogateway/web/guest/instrument/mipas/mipas-spectroscopic-database.

IG2 data set is temporarily available at the page:

https://earth.esa.int/eogateway/instruments/mipas/products-information?text=mipas, clikking on the link: Climatological IG2 profiles (IG2),

then it will be definitively hosted at https://www.ceda.ac.uk/

*Author contributions.* P. Raspollini, M. Ridolfi, L. Sgheri, M. Gai, B. M. Dinelli, F. Barbara, B. Carli and S. Ceccherini are the main authors

of the MIPAS/ESA ORM V8 processor. N. Zoppetti developed the statistical tools to filter, average, plot the MIPAS Level 2 V8 products,
M. Bianchini and S. Della Fera performed tests on Level 2 products. J. Remedios, H. Sembhi and D. Moore developed the Initial Guess
database for L2 processor (IG2), M. López-Puertas provided climatological profiles extending up to 200 km for the IG2, M. Chipperfield
provided diurnally varying SLIMCAT data set for the IG2, E. Arnone matched profiles from ECMWF ERA-Interim reanalysis to MIPAS
measurements. J-M Flaud compiled the MIPAS dedicated spectroscopic database pf4.45 and M. Ridolfi tested any improvements of this data

set with MIPAS observations. A. Dudhia provided the selected spectral intervals used for the analysis of each target species at the different
altitudes and the cross-section look-up tables, M. Kiefer performed tests to evaluate the impact of the improvements in the L1b processor on
L2 products. P. Raspollini was the scientific responsible of the ESA study aimed to finalise the ORM V8 for MIPAS/ENVISAT full mission
reprocessing, A. Dehn supervised the work as ESA technical officer, A. Piro performed checks on both L1 and L2 codes.

*Competing interests.* The authors declare that they have no competing interests.

*Acknowledgements.* This work was performed under ESA contract no. 4000112093/14/I-LG. We thank the MIPAS Quality Working Group
for the work done to improve both the Level 1 and the Level 2 processors.



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
