# Peer review of "Level 2 processor and auxiliary data for ESA Version 8 final full mission analysis of MIPAS measurements on ENVISAT"

_Atmospheric Measurement Techniques, 2021_

## Author Response (AR1)

**Level 2 processor and auxiliary data for ESA Version 8 final full mission analysis of MIPAS measurements on EN-VISAT, by Raspollini et al.**

Detailed point-by-point response to all referees' comments and list of all changes in the revised manuscript.

The response to the Referees has been structured as follows: (1) comments from Referees are reported in italics, (2) author's
5  response follows, (3) author's changes in manuscript, if not evident from author's response, are reported afterwards.

**1  Answer to the first referee**

**1.1  Main comment**

*My main comment concerns the treatment of horizontal inhomogeneities. A first order model is used, which only accounts for linear variations using a horizontal gradient. Statistically, improvements are achieved, e.g. with regard to day/night differences*
10  *(ascending/descending orbits). However, in situations of high (small scale) variability, e.g. due to wave breaking, using a linear gradient may even be counterproductive. For example, the authors state that closest coincidences of ERA-Interim ECMWF data for each MIPAS profile were used for computing the horizontal gradients of temperature, water vapour and ozone. How is the gradient then calculated? By simple linear interpolation? Since the horizontal resolution of the ERA-Interim data is higher than that of MIPAS (distance of the tangent points), the calculation of an effective gradient would make sense, which also*
15  *accounts for smaller-scale variability along the line of sight. If a simpler procedure was chosen, the authors could briefly discuss this and state that essentially a statistical (climatological) improvement was targeted.*

The inclusion of the Horizontal Gradient (HG) model in the ORM has been long debated, especially after Kiefer et al., 2010 found the so-called ascending descending (AX-DX) differences, correctly attributed to the fact that horizontal variability in the atmosphere along the line of sight was not taken into account. The AX-DX differences can be appreciated by calculating,
20  separately, for the AX and DX parts of the satellite orbit, the profile zonal means. Being of the order of the measurement noise error, the effect is hardly visible in the individual profiles.

The introduced HGs permit to model a linear horizontal variation of the atmospheric state parameters. While the linear approximation does not allow to account for the smaller-scale variability along the line of sight, still, it is better than the "constant" approximation corresponding to the horizontal homogeneity assumption. We agree that there may be situations (e.g.
25  due to wave breaking) in which the externally supplied HGs do not improve the capability of the forward model to reproduce the actual observation. These situations, however, should be regarded as "exceptions", because from the statistical point of view the quality of the fit turns out to be improved when HGs are modelled.

The dataset used for the calculation of the HGs influences the reduction of the AX-DX differences. In Figure 1 we compare, for different latitude bands, the temperature AX-DX differences obtained with HGs extracted from various sources. In this
30  case, the processed orbits refer to the measurements acquired in December 2007 (Optimized Resolution, OR mission phase).

The red curves refer to the AX-DX differences obtained by setting to zero the HGs. As we can see, the HGs extracted from the IG2 (Remedios et al, 2008) climatology provide only a marginal reduction of the AX-DX differences (grey curves) as compared to the cases of assuming HGs extracted from ECMWF (gold curves) or from a previous ORM processing with no HG model (blue curves). The HGs calculated from profiles retrieved in a previous ORM processing without gradients, and
35  HGs calculated from ECMWF profiles reduce the AX-DX differences by very similar amounts. This result points at the fact that HGs inferred from ECMWF, though computed from simple profile differences, actually act as "effective" HG estimates as those obtained from a previous ORM analysis assuming no horizontal variability. Conversely, the mere climatological estimate based on the IG2 database shows a worse performance. Note that, as expected, the temperature AX-DX differences do not vanish completely everywhere. Since the AX profiles are measured at night time, and the DX profiles at day time, the solar
40  tides are still visible in the differences.

Modifications in the revised paper: Added the following paragraph in Sec. 4.1.:

Since the horizontal resolution of MIPAS is of the order of the horizontal separation between the measured limb scans (Von Clarman et al., 2009), one may argue that atmospheric variabilities at smaller scales should contribute to the individual measurements with a signal smaller- or of the order of the noise. According to this reasoning, we tried to determine *effective*
45  HG estimates from the differences between profiles retrieved from adjacent MIPAS scans in earlier reprocessings (with the

[Figure]

**Figure 1.** AX-DX differences in various latitude bands (see plot's key) for the retrieved temperature profiles.

assumption of horizontal homogeneity). These HGs were actually found to reduce markedly the ascending / descending differences in the L2 products. Another estimate of HGs could be determined on the basis of the ECMWF profiles. We associated to each MIPAS limb scan the closest ECMWF profile in space and in time. HGs were then determined from the differences between ECMWF profiles associated to adjacent limb scans. The HGs computed with this approach were found to reduce the ascending / descending differences exactly as the HGs determined from previous MIPAS reprocessings. Considering that profiles retrieved in earlier MIPAS reprocessings (V7 and earlier) may sporadically show unphysical oscillations, for the V8 reprocessing we decided to use HGs determined on the basis of the ECMWF dataset. Specifically, from the ECMWF dataset we determined the HGs of Temperature, $H_2O$ and $O_3$. The HGs of the other gases were set to zero.

*Other comments:*

*l.11 Introduce IG2 (Level 2 Initial Guess). The abbreviation is used a few times, but not explained until l. 318.*

Done

*l.23 The effects are not limited to the region from the surface to the mesosphere. For example, solar activity also affects the thermosphere.*

Corrected

*l.26. I find the Kidston et al. citation a bit too specific here. I would therefore also use another citation, e.g., from the IPCC.* We have cited also 'Climate change 2021: the physical science basis, IPCC'

*l.27 A citation for the air quality aspect is missing.* We have added as an example the reference to this recent paper:

Mingcheng Wang, Qiang Fu, Stratosphere-Troposphere Exchange of Air Masses and Ozone Concentrations Based on Reanalyses and Observations, JGR Atmosphere, https://doi.org/10.1029/2021JD035159, 2021

*l.30 ... period (Fischer et al., 2008).* Done.

*l.33 Indicate approximately how large the resolution is.* We have added the following sentence: This is mainly driven by the step of the measurement tangent altitude grid and can reach for most trace species the values of 2-3 km in the altitude range 6-40 km, with better performances in the second phase of the mission (see Sec. **??**) where the measurement grid is finer.

*l.42 The statement does not apply to every tangent point since a global fit procedure is used?* Yes, indeed we write 'spectra', all the spectra of a limb scan are simulated and fitted at once.
In the revised paper we wrote: 'all atmospheric emission spectra of a limb scan'

*l.78 I would put the Fischer quote at the end of the sentence.* Done.

*l.85 A sketch of the measurement geometry would be helpful for some readers.* We have added a reference to Fig.5 of Fischer et al.

*l.88 What is meant by revisit time of three days? It should be about a month.* We meant that after three days we get an almost global coverage of MIPAS measurements, but it is true that 'revisit time' has a different meaning. In the revised paper we reworded the sentence as follows: 'The quasi-polar orbit allowed a global coverage, with MIPAS covering almost the whole globe in three days.'

*l.112 Both ... vary* Corrected

*l.130 I would say what is meant. Horizontal gradients of temperatures and trace gases.* Done

*l.140 ... and lower thermosphere?* Done

*l.143 I assume that this sequential procedure is also iterated?* No, the procedure is not iterated. The consecutive limb scans of an orbit are analysed sequentially. In general, the profiles retrieved from a given limb scan turn out to be a sufficiently good approximation of the atmosphere to be used as a background, to process the next scan. No modifications were performed in the revised paper.

*l.165 Briefly state the purpose.* We have added that IG2 are used 'for defining initial guess profiles, assumed profiles of the

interfering species, horizontal gradients'

*.171 Statistically, it shows an improvement.* We have reworded the text as follows: 'For most of the retrieved gases we observe a reduction in the V8 chi-square as compared to V7, indicating a better representation of the observations. This improvement is more significant in the OR phase of the mission, where the reduced spectral resolution makes more critical the interference of non target gases. The retrievals characterized by the largest reduction in the chi-square are $H_2O$ (-8%), $O_3$ (-20%), $HNO_3$ (-5%), $N_2O$ (-5%), $NO_2$ (-7%), $CFC-12$ (-6%), $N_2O_5$ (-9%), $ClONO_2$ (-14%). According to the results of dedicated sensitivity tests (not presented here), the obtained chi-square reduction is mainly due to the modifications in the building of the profiles of the gases that spectrally interfere with the target gas (see Sect. **??**). The improved spectroscopic data used in V8 (see Sect. **??**) and the introduced horizontal gradient model (see Sect. **??**) also contribute, albeit with a lesser extent, to the observed chi-square decrease. '

*l.193 mean retrieved profiles* Done

*l.221 in about » about* Corrected

*l.257 better "between observed and simulated spectral features".* Corrected

*l.258 consistency check is o.k., but the 7.6 μm region should not be further included in the HNO3 retrieval itself by just to account for HNO3 interferences in the 7.6 μm region.* The sentence refers to a separate study published in Perrin et. al 2016. For the $HNO_3$ V8 retrievals only microwindows in the 11 μm region are actually used (see http://www.atm.ox.ac.uk/group/mipas/err/). No modifications were made in the revised document.

*l.268 ... improves the spectral simulations...* corrected

*Legend Figure 10: Perhaps spell out FR and OR again.* Ok, done

l.331/332 The sentence is difficult to understand. It has been reworded as follows: 'Among the different databases, only the IG2 profiles are available on the altitude range from 0 to 120 km, the others being defined on a restricted altitude range. The extension of these profiles outside their native range is performed with the IG2 profiles which, in order to avoid discontinuities, are scaled by the ratio between the value of the profile at each edge of its native altitude range and the corresponding climatological value interpolated at the same altitude.'

*l.333 If necessary, give some details for priority system.* We have added a reference to the paper Dinelli et al., 2021.

*l.337 mb » hPa* Corrected

*l.357 Delete space after "range".* Done

*l.364 What do you mean by 0 hPa?* This is what is written in the paper that is cited in the paper. The "0 hPa" is the number quoted in table 1 of Chipperfield 2006 TOMCAT paper - https://rmets.onlinelibrary.wiley.com/doi/epdf/10.1256/qj.05.51. This value of "0 hPa" is based on the values from Briegleb, 1992. We have replaced "0 hPa" with "top-of-atmosphere".

*l.406 I don't understand this big difference between models levels and pressure levels right away.* This is how ERA-Interim data are provided. For data provided on pressure levels the lowest pressure is 1 hPa, for data on model levels the lowest pressure level is 0.1 hPa.

A small change in the text has been done for clarity reasons: Data on model levels were adopted instead of pressure levels in order to obtain a greater vertical coverage since they are released up to 0.1 hPa (about 65 km altitude) versus 1 hPa (about 50 km altitude), respectively. The adoption of levels with a fixed pressure scale implies an inhomogeneous upper limit of the profiles along the orbit.

*l.486 I would somewhat rewrite how you reduce outliers after the retrieval, e.g. ... after the retrieval based on a more sophisticated quality flag.* Actually this is only an introduction. Later, Sect. 6.2 describes the details of the procedures used to reduce the number of outliers in the products. No modifications in the revised paper were made

*l.620 long-term evolution* Done

*Figure 23: Improve quality of symbols and lines in the plot.* This figure has been replaced by another figure as suggested by

the second referee.

130     *l.655 ... observed scene multiplied...* The sentence has been reworded.

        *l.840 Remove the first parenthesis.* Done

**2   Answer to the second referee**

*L11 IG2 not defined* Now it is defined

        *L23 . . . could also mention pyroCb events arising from wildfires e.g. Australian New Year fires in 2020.* Done

135     *L23 [Taken together] these changes* Done

        *L29 "of the atmospheric emission" . . . perhaps you meant to say "of the infrared emission" otherwise better to just say "of the atmosphere"* We have changed with: 'of the infrared emission of the atmosphere'

        *L30 Which is correct . . . ENVISAT (as in the title and elsewhere) or Envisat?* We have replaced Envisat with ENVISAT all over the paper.

140     *L33 probably should jam "infrared" in this line before "emitting"* I have written '... constituents emitting in the middle infrared'

        *L34 10 years is a long time. . . CO2 increased quite a bit - how was that handled for the retrieval of temperature?* As mentioned in Sect. 4.3.1, CO2 (as well as CFC's) trends are accounted for in the IG2 climatology. Thus, temperature is retrieved assuming the CO2 profile valid at the time of the measurement. We have added the following sentences: 'For specific

145     gases (i.e. chlorofluorocarbons and $CO_2$), trends are accounted for in the IG2 climatology. Although in the IG2 database the profiles are tabulated only for a limited set of latitude bands and for each season of the years from 2002 to 2012, discontinuities in the L2 products are avoided using linear interpolation. Specifically, for each processed limb scan, the ORM V8 builds the corresponding IG2 profile estimates by linearly interpolating the IG2 tabulated profiles to the time and the geolocation of the considered measurement.'

150     *L67 I'm sure you have stories of "what not to do" that would be similarly helpful!* Yes, we agree, however a full book would be necessary to tell all of them...

        *L80 10 hour => 10:00* Corrected

        *L97 maximum [interferometetric] path ?* Added

        *L100 reducing the spectral resolution to* Yes, corrected.

155     We have written: '... corresponding to a FTS spectral resolution $\delta\sigma = 1/(2 \cdot \text{MOPD}) = 0.025 \text{ cm}^{-1}$' and: 'corresponding to a coarser FTS spectral resolution of $\delta\sigma = 0.0625 \text{ cm}^{-1}$.'

        *L107 MIPAS was not operating routinely for a long period. Since a 20 cm path was causing mechanical problems, what was the largest path thought to still be safe? Was a new max path limit determined from in-orbit testing?* You are right that our description seems to indicate that the changes in the settings of the instrument was decided to optimise the trade-off between

160     spectral and spatial resolutions, while the change was forced by mechanical problems, as also written at line 99. The new "safe" max path limit was determined from in-orbit testing.

        In the revised manuscript we have removed the sentence 'Having already obtained information on the spectral characterisation in the first part of the mission, for the long-term operations an improvement of both vertical and latitude resolution was considered to be a good compromise.'

165     The new text is: 'After the detection of this anomaly, on the basis of in-flight tests, a new *safe* value of 8 cm was established for the maximum interferometric optical difference (MOPD). With this MOPD, an unapodized spectral resolution

$\delta\sigma' = 0.0625$ cm$^{-1}$ is achieved, with a total time of 1.8 s required for the measurement of a limb spectrum. The savings in measurement time were then exploited both to implement a finer vertical sampling of the atmospheric limb, and to acquire additional limb scans within each orbit. Due to this optimised (more dense) spatial sampling, the measurements acquired from January 2005 onward are referred to as optimized resolution (OR) measurements.'

170

*L108 In the OR scan the lowest tangent height was increased from 6 to 7 km. Was that for an engineering mechanism reason or was it decided to trade-off the the 6-7 km region in order to add 4km in the mesosphere?* Actually, in order not to repeat

information reported in the paper of Dinelli et al., 2021 it was not said that, associated with the change of the retrieval vertical grid, this grid became floating, that is changing with latitude to better follow the tropopause height. Hence the lowest altitude

175 is 5 km in the polar region, 7.05 km at 45° and 12 at the equator.

This information is now included in the revised paper and a reference is added to the paper of Dinelli et al. 2021 as follows: 'The nominal FR (OR) scan pattern consists of 17 (27) sweeps with tangent heights in the range from 6 to 68  km (from 5-12 to 70-77 km according to latitude ) with 3 (1.5) km steps in the upper troposphere-lower stratosphere (UTLS) region. It should be noted that associated with the change of the measurement vertical grid occurred in the OR phase, this grid became floating,

180 i.e. the grid moved rigidly following the lowest tangent altitude determined at each latitude according to a latitude-dependent law to better follow the tropopause height and to have at least one sweep below the tropopause. Hence the covered altitude range is 5-70 km in the polar region, 7.05-72.05 km at 45° and 12-77 km at the equator. Further details are provided in (**?**).'

*L139 non-LTE* Done

*L145 What about the (approximate) reference height so you can vertically locate the limb scan. Where does that come from?*

185 *Do you have an internal a geopotential height product?* This is explained in Sect. 5.4 of Dinelli et al. 2021, https://doi.org/10.5194/amt-

2021-215, 2021. We have added the reference to this paper also here. We have added the sentence:

'Then the altitude grid is re-built starting from the lowest engineering tangent altitude corrected using information from co-located ECMWF altitude/pressure profiles (see Dinelli et al., 2021). '

*L210 Could you give some indication of the error incurred in assuming no horizontal gradient across the line-of-sight in*

190 *the refractive index? Also this is the only mention of the Curtis-Godson approximation (CGA) and should have a reference to the original work at least and possibly to other documented retrieval codes for ir limb sounders. What led to the choice of CGA and were any other alternatives considered e.g. Emissivity Growth Approximation (EGA) or Correlated-k (presumably not needed as the micro windows are quite narrow)?* A previous study (Ridolfi and Sgheri, AMT, 2014) assessed the error on

the retrieved tangent heights by assuming the standard 1976 US atmosphere versus the retrieved atmosphere. The error is due

195 to the difference in the calculated refractive index needed to solve the Eikonal equation to calculate the ray-tracing.

The error assessed is less than 200 m, and reaches these values only at the lowest tangent altitudes. Now, since the difference in the atmosphere introduced by the horizontal variability is much smaller than that considered in Ridolfi and Sgheri, 2014, the difference in the tangent altitudes due to the change in the refractive index is also much smaller.

The same paper of Ridolfi and Sgheri, 2014 is the source of the line-tracing algorithm implemented in the ORM V8. The

200 paper also compares the ray-tracing algorithm implemented in the ORM to other known methods.

In the revised manuscript we included the reference to the papers introducing the CG integrals and to the initial ORM paper were the all the adopted choices are analysed / justified.

*L211 What is the spacing of the angular grid used to represent the climatology along the line of sight?* In order to speed up

the computation of the CG integrals, the Eikonal equation is solved in Cartesian coordinates with respect to the arc parameter

205 $s$. In each layer the arc parameter increment ($ds$) is kept fixed. Each time we cross a layer we check the second derivatives of the integrand function, in order to define the new $ds$.

In fact, due to the horizontally changing atmosphere, the CG are really line integrals, that can be transformed into simple Riemann sums if the increment $ds$ is constant. Also, $ds$ should change according to the curvature of the line of sight, with a smaller $ds$ in the lower layers where the refraction is more relevant.

210 Tests have been performed to set the $ds$, balancing speed and accuracy in the calculation of the CG integrals.

In the modified manuscript we better explain how ECMWF profiles are used for generating the HG: 'HGs were then determined from the differences between ECMWF profiles associated to adjacent limb scans.'

*L228 Do you really have the horizontal resolution to achieve a 2nd order correction?* Please check also the related answer

to the first referee. The correction on the single retrieval is not statistically significant, so we do not claim that we reproduce a 2nd order effect on the 2D atmosphere.

The improvement can be seen as the reduction on the AX-DX differences, calculated on latitude band averages.

However, the reduction is not only due to a climatological effect. Values of the horizontal gradients calculated from the actual profile values (either from ECMWF or from a previous MIPAS reprocessing) have to be inserted in the model to improve the reduction. On the other hand we do use some simplifications, such as neglecting the fine-scale inter-scan profile variations, and assuming that the gradient is constant in each interval defined by the scan pattern. See also reply to first referee. A sentence has been added at the end of Sect. 4.1: 'The remaining seasonality in latitude band averages of V8 measurements could be due to the fact that our model of the horizontal variability of the atmosphere is a first order model with a gradient, i.e. we only model an inter-scan linear variation of the atmospheric state. Un-modelled smaller-scale atmospheric variabilities, while contributing with a signal below the noise in the individual measurements, could cause a visible effect in the averages.'

*L342 What is the effect on temperature on the discontinuous CO2 concentration change on crossing a year boundary?* Within

the IG2 climatology, profiles like $CO_2$ and CFCs are corrected for their trend also seasonally, in addition the profiles are varied continuously by linearly interpolating with time the IG2 profiles relative to each season. As a consequence, no discontinuity is present in $CO_2$ profiles. We have added a sentence, as already described for a similar comment of the first referee.

*L455 How about a joint retrieval of H2O and the HDO/H2O ratio?* Thanks for the suggestion, this should be tried indeed.

Actually, the microwindows have been defined in order to reduce the interference between $H_2O$ and HDO lines and, thus, the correlation between the retrieved profiles. However, we have found that the use of retrieved $H_2O$ profile as a priori for the HDO profile may introduce some bias in the averaged HDO profiles, and we are using a dedicated approach to reduce the bias (see Ceccherini et al., 2014) . No modifications were performed in the modified manuscript.

*L454 Where does the magic number 31.6 come from? Need some context here e.g. over what values does the continuum range?* The a priori error for the offset was defined by looking at the typical variability of the offset retrieved for the MWs

relating to gases for which optimal estimation is not used. A reasonable value for the a-priori error is obtained by putting a value of 1000 on the diagonal of the a priori covariance matrix. The value of 31.6 corresponds to the square root of 1000. We have modified the text as follows: '...31.6 $nW/(cm^2 * sr * cm^{-1})$ (equal to the square root of 1000, defined as the typical variability for the offset value retrieved for each microwindow)'

*L470 What is the size of the matrices that need to be furnished to accomplish this feat. Would representative matrices suffice i.e. if they be aggregated over seasons and latitude bands?* For this analysis it is sufficient to provide the a priori profile,

the covariance matrix and the averaging kernel matrix for each retrieved profile. These quantities are already provided in the current output files. The aggregation of the data can be decided by the user, no other information has to be provided to perform this analysis. In the revised manuscript we have specified that the only quantities that are needed are the used a priori, the products and the quantities that characterize them (CM and AK), all these quantities are provided in the MIPAS output files.

'This new method removes the downsides of the use of a priori information, provided that information about the used a priori is made available to the users together with the products and the quantities that characterize them (CM and AK).'

*L593 and L683 not-good -> bad I suppose* Yes, changed

*L594 So are the VMR retrievals also marked as "bad" or some other specific flag that indicates missing data because the temperature retrieval failed?*

No profiles are reported in this case, because retrieval was not performed. No specific flag are available indicating the missing data, but it is possible to reconstruct the cause of the missing data by looking at the flag of temperature profile. We have added the following sentence: 'the cause of the missing VMR profile in the output files can be deduced by the temperature profile flag.'

*L603 and Page 33 Fig 22 Representative retrieval [product]? What product and OR or FR? If this distribution appears (or not) in Fig 1 or 2 then please indicate which one.* We have specified that the distribution refers to p-T retrieval of FR phase as

follows: 'Figure **??** shows as an example the chi-square distribution of p-T V7 retrievals relative to the FR phase'

*L608 the values of the thresholds [for each product]* Corrected.

[Figure]

**Figure 2.** Temperature differences retrieved from V8 and V5 L1b files for the 4 pressures indicated in the key as a function of time

*L601 derive [atmospheric] trends* Done

*L636 and Fig 23 Fig 23 is not very good for showing the temporal variation. The sequence of colors/symbols bears no relation to the time domain. I suggest taking only three pressure levels for UTLS, LS and MS and plotting the Tdiffs vs time. Also then you could indicate where the decontaminations (L653) occur on this figure.* We have replaced Fig.23 with the new

suggested figure reported below. In the new figure We have not indicated the decontamination periods because we think that this piece of information is not useful in this plot. The selected orbits used for this test were chosen far from the decontamination periods in order to avoid problems connected with particular situations.

*L686 Some indication of file sizes would be useful*

We have added the sentence: 'The size of the files varies with species and observation mode, since it depends on the number of fitted parameters. On average the size of a standard file is about 1 MB, while that of a extended file is about 3 MB.'

*L698 If 2nd order effects are a big deal (I doubt that the auxiliary model data can be used to correct to this order anyway) then you should indicate that these gradients are limited to a linear assumption along the line of sight.* Yes, we agree with the

reviewer, in the revised manuscript we have modified this paragraph.

The new paragraph is: 'The L2 processor ORM V8 and its auxiliary data are significantly different as compared to the ESA L2 processor ML2PP V7. The radiative transfer model was improved by allowing for a linear variation of the atmospheric state (with a gradient) to approximate the actual horizontal variability of the atmosphere sounded by the instrument line of sight. Horizontal gradients computed from ECMWF ERA-Interim reanalysis were assumed in the ORM V8 processing. This improved model implied a significant reduction of the ascending-descending differences that, initially, were erroneously attributed to day / night differences in profiles that actually, are not affected by a diurnal variability.'

**3 Other changes**

The caption of Fig. 14 has been made homogeneized to other figures

Corrected reference to two papers (Dinelli et al., 2021 and Pettinari et al., 2021) which were published in the meantime. Added acknowledgements for the use of ACE data in the compilation of IG2 dataset.

---

## Author Response (AR2)

**Reply to the editor**:

Thanks for these corrections. The editor's comments are reported in italics before each authors' reply.

*L90: Fig.5 --> Fig. 5*

Done

*L153: pT retrieval, suggest to write p,T in italics to highlight that these are physical symbols*

Done

*L184-185: makes more critical the interference --> word order is a bit awkward. How about: 'makes the interference ... gases more critical'*

Done

*L308: suggest to write 'we refer' rather than 'please refer'*

*Done*

*L323: 50pptv --> 50 pptv*

Actually, I have found '50 pptv .' I have corrected it in '50 pptv.'

*L379: one too many bracket after NH3?*

No, the first bracket is for defining the two references of the GEOS-Chem model profiles, the second bracket is for defining the second reference (Shepard, private communication). I have changed the position of 'for NH3' to make the text clearer:

… GEOS-Chem model profiles (Bey et al. (2001) and, for $NH_3$, Shephard (private communication))

*L412, 427-428: please write 10:00 and 22:00 to stay consistent with your time indication throughout the manuscript.*

Thanks for spotting these inconsistencies. I have preferred to use the 10 am/pm time indication throughout the manuscript.

*L429: after 90 there should be a degree symbol I guess*

*Ok, added*

*L486: why nw/(cm2 sr cm-1) is printed in italics?*

Corrected